# UMB: Understanding Model Behavior for Open-World Object Detection

**Xing Xi**      **Yangyang Huang**      **Zhijie Zhong**      **Ronghua Luo**[*]
School of Computer Science and Engineering
South China University of Technology
GuangZhou, China 510006

## Abstract

Open-World Object Detection (OWOD) is a challenging task that requires the detector to identify unlabeled objects and continuously demands the detector to learn new knowledge based on existing ones. Existing methods primarily focus on recalling unknown objects, neglecting to explore the reasons behind them. This paper aims to understand the model's behavior in predicting the unknown category. First, we model the text attribute and the positive sample probability, obtaining their empirical probability, which can be seen as the detector's estimation of the likelihood of the target with certain known attributes being predicted as the foreground. Then, we jointly decide whether the current object should be categorized in the unknown category based on the empirical, the in-distribution, and the out-of-distribution probability. Finally, based on the decision-making process, we can infer the similarity of an unknown object to known classes and identify the attribute with the most significant impact on the decision-making process. This additional information can help us understand the behavior of the model's prediction in the unknown class. The evaluation results on the Real-World Object Detection (RWD) benchmark, which consists of five real-world application datasets, show that we surpassed the previous state-of-the-art (SOTA) with an absolute gain of 5.3 mAP for unknown classes, reaching 20.5 mAP. Our code is available at https://github.com/xxyzll/UMB.

## 1 Introduction

As a fundamental task in computer vision, object detection has always been the focus of extensive attention[1, 2, 3]. Traditional object detection methods are trained on closed datasets, assuming all detected objects have already been annotated in the training set. However, the real-world environment's complexity means it is impossible to annotate all objects. As a result, the application of traditional detection methods is limited. Open World Object Detection (OWOD) has been introduced to address the issue. OWOD can be divided into two subtasks: mining potential objects and incremental learning. The former requires the model to detect categories in the test set that have not been annotated in the training set. These newly discovered objects are then handed over to annotators, who select the categories of interest. Subsequently, the model is required to fine-tune its existing knowledge to detect these newly added categories (incremental learning).

Existing works primarily focus on generating pseudo-labels for potential objects in the training set, treating these pseudo-labels as annotations for unknown categories. For instance, ORE[4] labels samples with high objectness predicted as background as potential objects. CAT[5] and RE-OWOD[6] utilize selective search to provide annotations for unknown categories. OW-DETR[7] proposes an attention-driven pseudo-label strategy to mine potential positive samples. However, despite these

---

[*]Corresponding author: `rhluo@scut.edu.cn`.

38th Conference on Neural Information Processing Systems (NeurIPS 2024).

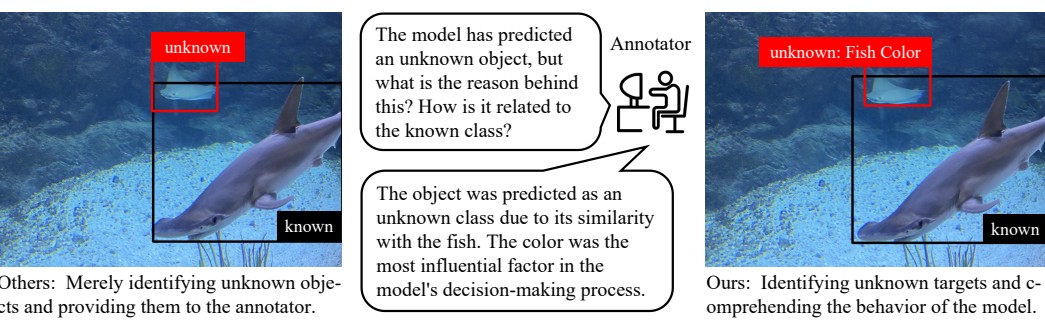

Figure 1: An illustration of our UMB and other methods. Previous OWOD methods only detected unknown objects (left), while our method further understands the model's behaviour (right).

heuristic methods being able to recall potential objects, they share a common flaw. As shown in Figure 1 (left), existing methods can only detect unknown objects and then provide these labels to annotators. However, the reason the model would predict these objects remains unknown to the annotators. Therefore, this paper attempts to understand the model's prediction of potential objects, establish connections between unknown and known categories, and then provide this additional information to annotators.

To achieve this, we propose a novel model (UMB) that uses textual attributes to mine potential unknown objects. Specifically, we first define targets that share similar attributes with known categories but are predicted as background as potential objects. Then, to find these potential objects, we build a distribution that associates attribute similarity with the probability of positive samples, which can be seen as the empirical probability of an object possessing a particular attribute being classified as a positive sample. If a sample predicted as background has a high empirical probability and attribute similarity, we regard it as an unknown object. Finally, based on the decision-making process, we infer the most similar known classes with the unknown object and calculate the most significant impact attribute. As shown in Figure 1 (right), our method can identify the unknown and provide information about their connections with known categories and the influence of attributes on decision-making.

We evaluated our method on the Real-World Object Detection (RWD) benchmark composed of datasets from five practical applications, and our method achieved significant improvements. We improved almost all datasets, surpassing the OVC (GT) that uses real class names. Significantly, in the Surgery[36] dataset, we achieved the **213%** performance in unknown category. The main contributions of this paper are as follows:

- To the best of our knowledge, we are the first to notice the limitations of models on unknown predictions and attempt to understand the predictive behaviour of models.
- To achieve this, we propose a new model framework (UMB) that can detect unknown categories and understand model behavior utilizing the textual description of known categories.
- We model the textual attributes and the probability of positive samples to obtain the empirical probability. By combining the empirical probability, the in-distribution probability, and the out-of-distribution probability, we are able to discover unknown categories.
- The evaluation results on the RWD benchmark show that our method achieved significant performance improvements, establishing a new state-of-the-art (SOTA) with 5.3 mAP advantages in both known and unknown category performance.

## 2 Related Works

### 2.1 Open Vocabulary Object Detection

Open Vocabulary Object Detection (OVD), as a subset of open-world perception, was initially introduced by OVR-CNN[8]. OVD employs the text encoder to transform classes needing detection

into text embeddings, determining the current object's class by calculating the similarity between all text and visual embeddings. Subsequent works further have enhanced OVD's performance, including knowledge transfer from pre-trained models through distillation[9, 10, 11], the addition of high-quality object candidates[12, 13], and alignment of text-visual regions[14, 15, 16, 17, 18]. However, in the setting of OWOD, these methods fail to detect the unknown class due to the uncertainty of object categories. Our approach, based on OWL-ViT[19], broadens OVD to OWOD by modelling the correlation between text attributes and the probability of positive samples.

## 2.2 Open-World Object Detection

Open World Object Detection (OWOD), distinct from OVD, presents stricter settings and is a more challenging task, as proposed by ORE[4]. OWOD requires the detector not only to detect potential unknown objects without any information of unknown classes (including category names) but also to fine-tune the detector on newly introduced classes for continuous learning of new knowledge. Existing research focuses on heuristic assumptions for potential targets. ORE considers background samples with high objectness in RPN as potential unknowns, OW-DETR[7] calculates the average score of feature regions to determine positive samples, and PROB[20] proposes the use of Mahalanobis distance to discover the potential positive samples. Some other methods use additional pseudo-label generation mechanisms to generate annotations for potential objects, including selective search[5, 21, 22], random sample generation[23], and large model knowledge transfer [24, 25, 26]. However, these OWOD methods focus on detecting potential objects and ignore investigation into underlying reasons. Our method attempts to understand the behaviour of the model's unknown prediction, establishing a relationship between unknown objects and known classes.

# 3 Our Approach

Our method, named UMB, is built upon OWL-ViT[19], with the overall process illustrated in Figure2. First, what characteristics should of an unknown target possess? We posit that if an object is predicted as background but exhibits attributes of known classes, it should be considered an unknown target. Therefore, we model the attributes of known classes and the probability of positive samples to build distribution of the empirical probability (Sec. 3.2). The distribution represents the detector's empirical confidence in predicting objects with known class attributes as positive samples. If a background sample's empirical confidence and similarity to known class attributes (In-Distribution Probability Sec. 3.3) are both high, we consider it a potential object.

Then, since predictions for known and unknown classes are based on text attributes, we can infer the most similar known class based on the attribute similarity of the unknown object (eqn. 16). Finally, we can calculate the contribution of each attribute based on the decision-making process of unknown predictions, thereby identifying the attributes that have the greatest impact on decision-making (eqn. 17). This additional information can aid in understanding the model's behaviour in unknown classes.

## 3.1 Background

To obtain text that describe objects, we use the following template[28, 29] to request the Large Language Model (LLM) to list all attributes related to known classes:

$$Template(C, Z) = I \; am \; using \; a \; language - vision \; model \; to \; identify \; \{C\}. \; List \; the \\ \{Z\} \; attributes \; of \; \{C\}, \; which \; will \; be \; used \; for \; detection. \tag{1}$$

Where $C$ and $Z$ denote the class name and predefined attribute type (e.g., shape), respectively. These attributes are filled into the prompt template[27]:

$$Prompt(Z, A) = object \; which \; (is/has/etc) < Z > is \; < A >, \tag{2}$$

where $A$ is the attribute text generated in eqn. 1, e.g., blue. Then, those prompts are fed into the trained text encoder to generate text attribute embeddings $E_{att} = [e_{att_1}, e_{att_2}, ..., e_{att_n}]^\top \in \mathbb{R}^{n \times d}$, where d is the hidden dimension, e.g, 512. $n$ denotes the number of the text attributes. The image is fed into the trained visual encoder to generate visual embeddings $E_{vis} = [e_{vis_1}, e_{vis_2}, ..., e_{vis_m}]^\top \in \mathbb{R}^{m \times d}$, where $m$ represents the number of patch. In order to establish a connection between these class-agnostic attributes and known categories, we use additional weights $W \in \mathbb{R}^{m \times n}$ trained in known

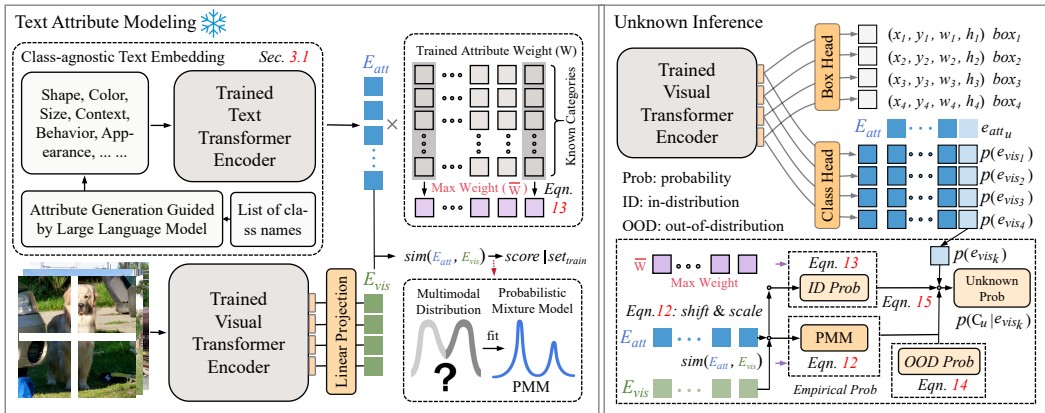

Figure 2: Overall structure of our UMB. It begins by populating prompt template with known class names and employing large language model (LLM) to generate attributes (Sec. 3.1). These attributes are then filled into template and encoded by text encoder to generate attribute embeddings ($E_{att}$). We model the attributes and their corresponding positive sample probabilities to build empirical probability (Sec. 3.2). We utilize the empirical, in-distribution and out-of-distribution probability to ascertain whether an object pertains to an unknown category (Sec. 3.3).

categories to linearly combine similarities. Therefore, given a visual embedding $e_{vis_i}$, the probability of its corresponding known category $j$ is:

$$
\begin{aligned}
p\left(C_j \mid e_{vis_i}\right) &= Sigmoid\left(w_{j,1} \cdot sim\left(e_{vis_i}, e_{att_1}\right) + \ldots + w_{j,n} \cdot sim\left(e_{vis_i}, e_{att_n}\right)\right) \\
&= Sigmoid\left(\sum_{k=1}^{n} w_{j,k} \cdot sim(e_{vis_i}, e_{att_k})\right),
\end{aligned}
\tag{3}
$$

where $Sigmoid$ is the Sigmoid activation function, and $sim$ denotes the Cosine Similarity. The pseudocode for known class prediction and attribute generation can be found in Algorithm 1.

## 3.2 Text Attribute Modeling (TAM)

### 3.2.1 Attribute Modeling

We model the attribute similarities in the training set with category confidence as the positive sample probability to build an empirical probability distribution. However, as shown in eqn. 3, the score is the linear combination of all attribute similarities, so it is influenced by all attributes simultaneously. Thus, we weigh confidence with linear combination weights ($W \in \mathbb{R}^{m \times n}$) to balance the contributions of different attributes. Specifically, given visual embedding $e_{vis_k}$, positive sample probability of attribute $i$ for category $j$ can be represented as:

$$
\tilde{p}\left(e_{att_i}, C_j \mid e_{vis_k}\right) = w_{j,i}^{1-\beta} \cdot p\left(C_j \mid e_{vis_k}\right)^{\beta}, \quad w_{j,i} = W[j,i]
\tag{4}
$$

where $\beta$ is a hyperparameter used to balance the contributions of weights and scores. For simplicity, we use the geometric weighted average. We incorporate all similarities in the training set and their corresponding probabilities of positive samples, establishing a mapping $f_{i,j} : sim(e_{att_i}, e_{vis_k}) \rightarrow \tilde{p}\left(e_{att_i}, C_j \mid e_{vis_k}\right)$. However, during training, the model cannot utilize the annotations of any unknown classes. Therefore, we define the positive sample probability of attribute $i$ for the unknown class as the maximum of its probability to all known classes. Specifically, the probability of attribute $i$ for the unknown class $C_u$ can be represented as:

$$
\begin{aligned}
\tilde{p}\left(e_{att_i}, C_u \mid e_{vis_k}\right) &= max\left(\tilde{p}\left(e_{att_i}, C_1 \mid e_{vis_k}\right), \ldots, \tilde{p}\left(e_{att_i}, C_m \mid e_{vis_k}\right)\right) \\
&= \underset{j \in [1,m]}{argmax}\left(\tilde{p}\left(e_{att_i}, C_j \mid e_{vis_k}\right)\right)
\end{aligned}
\tag{5}
$$

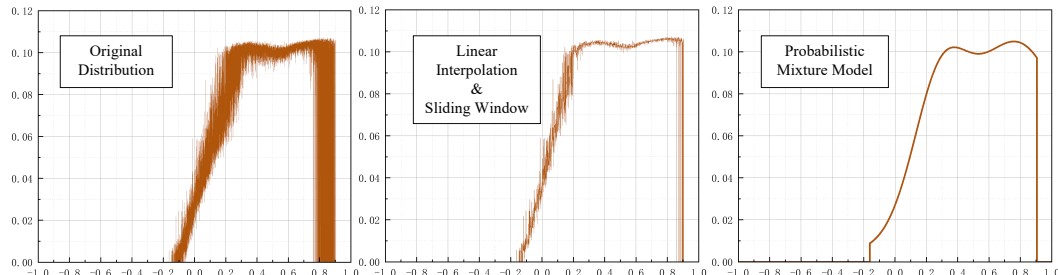

Figure 3: An illustration of the Probability Mixture Model. To establish a continuous probability distribution, we use linear interpolation on the original distribution (left) to estimate missing points and employ the sliding window to eliminate noise within the distribution (middle). Finally, we use the probabilistic mixture model to fit the optimized distribution (right).

### 3.2.2 Distribution Optimization and Fitting

To establish a continuous probability distribution, we need to optimize and fit the original distribution. First, contrary to the OWOD benchmark that heavily relies on extensive COCO[30] and VOC[31] data, RWD pays more attention to real-world application and is specifically designed for the few-shot setting. This results in the model not having sufficient samples to establish the probability distribution. Consequently, there are some undefined points in the mapping function $f_{i,j}$. To mitigate this, we employ the linear interpolation to estimate the values of these missing points $\underline{x}$:

$$f_{i,j}(\underline{x}) = k(\underline{x} - \underline{x}_l) + f_{i,j}(\underline{x}_l), \quad k = (f_{i,j}(\underline{x}_r) - f_{i,j}(\underline{x}_l))/(\underline{x}_r - \underline{x}_l), \tag{6}$$

where, $\underline{x}_l$ and $\underline{x}_r$ respectively represent the points to the left and right of $\underline{x}$ that are closest in the mapping $f_{i,j}$.

Then, we utilize the sliding window to filter the noise present in the distribution. With a predetermined window size, we calculate the maximum positive sample probability across the entire window to substitute the current value:

$$f_{i,j}(sim(e_{vis_k}, e_{att_i})) = \underset{a \in [0, W_{sz} - 1]}{argmax} f_{i,j}(sim(e_{vis_k}, e_{att_i}) + a), \tag{7}$$

where $W_{sz}$ denotes the window size. As depicted in Figure 3, the employment of linear interpolation and the sliding window ensures the original shape of the probability distribution remains intact, concurrently minimizing the noise inherent in the distribution.

Finally, as shown in Figure 3 (middle), the optimized probability distribution $f_{i,u}$ demonstrates the multi-peak characteristic. Consequently, we postulate that the original distribution is composed of multiple basic probability distributions (e.g., Gaussian Distribution). As a result, we employ the mixture probability distribution to fit the initial distribution. Specifically, we construct the model using the linear combination of multiple Gaussian distributions:

$$f_{i,u}(sim(e_{vis_k}, e_{att_i})) = \sum_{a=1}^{A} Gm(sim(e_{vis_k}, e_{att_i})|w_a, \sigma_a, \mu_a),$$
$$Gm(x|w, \sigma, \mu) = w \cdot \frac{1}{\sigma\sqrt{2\pi}} e^{\frac{(x-\mu)^2}{2\sigma^2}}, \tag{8}$$

where $A$ is the number of the Gaussian distribution. Additionally, we observed that certain attributes demonstrate the skewed distribution, suggesting that fitting with the Gaussian model may not be the optimal choice. Consequently, we utilize the asymmetric Weibull distribution as a substitute for the Gaussian distribution:

$$Wb(x|w, \lambda, k) = w \cdot \frac{k}{\lambda} \left(\frac{x}{\lambda}\right)^{(k-1)} e^{-(\frac{x}{\lambda})^k}. \tag{9}$$

In order to ascertain the parameters of these distributions, we designate them as learnable parameters and employ Mean Squared Error (MSE) as the loss function for optimization. The pseudocode for text attribute modeling can be found in Algorithm 2.

## 3.3 Unknown Inference

Following FOMO[27], we calculate the weighted mean of all attribute embeddings as the embedding for the unknown class:

$$e_{att_u} = \frac{1}{m} \sum_{j=1}^{m} \left( \sum_{i=1}^{n} e_{att_i} \cdot w_{j,i} \right) \in \mathbb{R}^d. \tag{10}$$

Following this, we utilize the pre-trained $scale$ layer ($\mathbb{R}^d \to \mathbb{R}^1$) and $shift$ layer ($\mathbb{R}^d \to \mathbb{R}^1$) for the purpose of scaling the similarity[19]:

$$T(sim(e_{vis_k}, e_{att_u})) = (sim(e_{vis_k}, e_{att_u}) + shift(e_{vis_k})) \cdot scale(e_{vis_k}). \tag{11}$$

Finally, we adjust the similarity of the average embedding. This adjustment is segmented into three components: empirical probability, in-distribution probability, and out-of-distribution probability.

**Empirical Probability (Empirical Prob)**. For known categories, each attribute contributes unevenly to the category score (eqn. 6). Hence, for the unknown class, merely using the summation of the empirical probability to ascertain category confidence is suboptimal. We utilize the maximum weight from the known class to balance the contributions from various attributes of the unknown class. Specifically, for the visual embedding $e_{vis_k}$, its corresponding empirical probability is:

$$\hat{f}_u(e_{vis_k}) = \sum_{i=1}^{n} f_{i,u}(sim(e_{vis_k}, e_{att_i})) \cdot \bar{w}_i, \ \bar{w}_i = \underset{j \in [1,m]}{argmax} \ w_{j,i}. \tag{12}$$

Herein, $f_{i,u}$ denotes the positive sample probability of attribute $i$ towards the unknown class, as established earlier.

**In-Distribution Probability (ID Prob)**. We aspire for the model to observe the known attribute of the current object. Consequently, we incorporate the weighted sum of the scaled attribute similarities:

$$f_{\text{ID}}(e_{vis_k}) = \sum_{i=1}^{n} Sigmoid(T(e_{vis_k}, e_{att_i})) \cdot \bar{w}_i. \tag{13}$$

**Out-of-Distribution Probability (OOD Prob)**. Both empirical probability and in-distribution probability are based on the model's prediction on known classes. Therefore, inevitably, the model predicts high empirical probabilities and in-distribution probabilities for known categories. To counteract this, we employ out-of-distribution probability to offset their influence:

$$f_{OOD}(e_{vis_k}) = \underset{j \in [1,m]}{argmax}(1 - \underset{j \in [1,m]}{Softmax}(T(sim(e_{vis_k}, e_{att_i})) \cdot w_{j,i})). \tag{14}$$

Ultimately, given the visual embedding $e_{vis_k}$, the corresponding confidence for the unknown class is denoted as:

$$p(C_u|e_{vis_k}) = Sigmoid((\underbrace{\hat{f}_u(e_{vis_k}) \cdot (1 - \alpha)}_{Empirical \ Prob} + \underbrace{f_{\text{ID}}(e_{vis_k}) \cdot \alpha)}_{ID \ Prob} \cdot \underbrace{f_{OOD}(e_{vis_k})}_{OOD \ Prob})$$
$$\cdot Sigmoid(\underbrace{T(sim(e_{vis_k}, e_{att_u}))}_{Average \ Similarity})), \tag{15}$$

where $\alpha$ is used to balance the contribution form in-distribution and empirical probability.

## 3.4 Additional Information

**Similarity between known and unknown classes**. Predictions for both unknown and known classes are determined by attribute similarity. Hence, we can compute its similarity with known classes based on the visual embedding of objects classified as unknown. Similar to unknown inference, we take into account the empirical probability of the current object and its confidence in being predicted as a known class. Specifically, for the visual embedding $e_{vis_k}$ of objects predicted as unknown, the corresponding similarity to known classes is:

$$S_u(e_{vis_k}) = \underset{j \in [1,m]}{softmax} \left( \sum_{i=1}^{n} f_{i,j}(sim(e_{vis_k}, e_{att_i})) + p(C_j|e_{vis_k}) \right) \tag{16}$$

| Task IDs(->) | Aquatic Task1 | | Aquatic Task2 | | Aerial Task1 | | Aerial Task2 | | Game Task1 | | Game Task2 | | Medical Task1 | | Medical Task2 | | Surgery Task1 | | Surgery Task2 | | Overall Task1 | | Overall Task2 | |
|---|---|---|---|---|---|---|---|---|---|---|---|---|---|---|---|---|---|---|---|---|---|---|---|---|
| | U | K | PK | CK | U | K | PK | CK | U | K | PK | CK | U | K | PK | CK | U | K | PK | CK | U | K | PK | CK |
| Base+GT-B | 29.8 | 45.0 | 45.0 | 36.7 | 1.3 | 5.7 | 5.7 | 1.4 | 15.0 | 0.4 | 0.4 | 0.1 | 0.5 | 0.0 | 0.0 | 0.1 | 5.6 | 1.5 | 1.4 | 0.3 | 10.4 | 10.5 | 10.5 | 7.7 |
| Base-FS-B | 7.1 | 41.1 | 41.1 | 31.9 | 1.2 | 10.4 | 10.1 | 4.0 | 16.0 | 4.6 | 4.8 | 3.9 | 0.6 | 6.1 | 6.1 | 3.3 | 1.3 | 11.9 | 11.3 | 10.9 | 5.2 | 14.8 | 14.7 | 10.8 |
| FOMO-B | 3.5 | 43.8 | 44.1 | 40.8 | 0.9 | 12.0 | 12.6 | 5.4 | 13.3 | 3.8 | 4.4 | 4.1 | 2.1 | 6.4 | 5.5 | 11.5 | 6.1 | 12.7 | 12.9 | 11.0 | 5.2 | 15.7 | 15.9 | 14.6 |
| Base+GT-L | 34.8 | 36.0 | 36.0 | 42.3 | 1.0 | 7.9 | 7.2 | 0.8 | 12.4 | 0.9 | 0.8 | 0.3 | 2.4 | 0.2 | 0.2 | 0.3 | 2.4 | 0.2 | 2.6 | 1.3 | 10.6 | 9.0 | 9.4 | 9.0 |
| Base-FS-L | 2.4 | 43.6 | 42.9 | 42.8 | 9.7 | 23.7 | 21.9 | 13.0 | 8.2 | 10.4 | 10.2 | 13.4 | 1.1 | 23.2 | 21.7 | 24.2 | 3.6 | 26.0 | 25.0 | 7.4 | 5.0 | 25.4 | 24.3 | 20.2 |
| FOMO-L | 18.2 | 50.1 | 48.1 | 47.1 | 6.0 | 25.3 | 23.7 | 16.0 | 30.4 | 10.7 | 9.9 | 11.2 | 9.4 | 21.8 | 19.9 | 34.6 | 12.0 | 29.0 | 28.9 | 8.5 | 15.2 | 27.4 | 26.1 | 23.5 |
| Ours: | | | | | | | | | | | | | | | | | | | | | | | | |
| UMB-Gm-B | 13.3 | 43.8 | 43.0 | 39.7 | 1.5 | 18.8 | 19.0 | 5.9 | 15.2 | 4.1 | 4.7 | 4.3 | 2.3 | 5.4 | 3.5 | 11.8 | 10.1 | 13.9 | 14.3 | 11.1 | 8.5 | 17.2 | 16.9 | 14.6 |
| UMB-Wb-B | 13.5 | 43.8 | 43.0 | 39.7 | 1.4 | 18.8 | 19.0 | 5.9 | 16.3 | 4.1 | 4.7 | 4.3 | 2.3 | 5.4 | 3.5 | 11.8 | 14.5 | 13.9 | 14.3 | 11.1 | 9.6 | 17.2 | 16.9 | 14.6 |
| UMB-Gm-L | 18.6 | 50.7 | 50.5 | 50.4 | 11.2 | 42.7 | 40.4 | 22.6 | 35.1 | 11.1 | 10.7 | 10.5 | 13.2 | 22.2 | 19.1 | 34.5 | 24.5 | 36.6 | 39.0 | 17.4 | 20.5 | 32.7 | 31.9 | 27.1 |
| UMB-Wb-L | 18.6 | 50.8 | 50.5 | 50.4 | 11.1 | 42.8 | 40.4 | 22.5 | 32.7 | 11.1 | 10.7 | 10.5 | 8.6 | 22.3 | 17.3 | 33.2 | 25.6 | 36.6 | 39.0 | 17.4 | 19.3 | 32.7 | 31.6 | 26.8 |

Table 1: Comparison with previous SOTA methods on the RWD benchmark. Base+GT represents the standard OVC setting using all class names including unknown label. Base-FS indicates the baseline of fine-tuning the benchmark model with the same supervision received[27]. B and L respectively represent two different sizes of the OWL-ViT model, B/14 and L/14. U, K, PK, and CK respectively represent unknown categories, known categories, previously known categories, and currently introduced categories. Overall indicates the average performance of the model on 5 datasets. Wb and Gm respectively represent use of Weibull and Gaussian distribution during the fitting stage.

**Maximal attribute contribution**. Attributes are used to compute the similarity with visual embeddings, and then the model makes predictions based on this similarity. Therefore, the contribution of each similarity can be calculated to determine the impact of a particular attribute in the decision-making process. For a visual embedding $e_{vis_k}$ that is predicted as an unknown class, the influence of attribute $i$ on the current decision is denoted as:

$$Ctr(e_{att_i}) = \overline{w}_i \cdot (Sigmoid(T(e_{vis_k}, e_{att_i})) \cdot \alpha + f_{i,u}(sim(e_{vis_k}, e_{att_i})) \cdot (1 - \alpha)) \qquad (17)$$

# 4 Experiments

## 4.1 More Details and Experiments

In our supplemental material, we provide detailed information about our experiments, including: comprehensive descriptions of the datasets (sec A.2), definition of OWOD (sec A.1), evaluation metrics (sec A.3), details (sec A.4), more extensive ablation studies (sec A.5), analysis and visualization of PMM training (sec A.6), similarity evaluation(sec A.7), attribute study (sec A.8), discussion of the limitations (sec A.10) and broad impact(sec A.9).

## 4.2 Datasets

The OWOD benchmark is established on the VOC[31] and COCO[30] datasets. In the era of foundation models, the zero-shot capability of detectors on such datasets has reached its limit, for instance, OWL-ViT[19] unknown recall is 79.0. Therefore, following FOMO, we have shifted the benchmark for evaluating detector performance to the more practically applicable RWD benchmark.

## 4.3 Comparison with Other State-of-the-art Models

Table 1 presents the comparison of our UMB method and previous SOTA methods established on the RWD benchmark. Overall, our method achieved comprehensive leadership, surpassing previous methods with the unknown performance advantage of **4.4** mAP (Wb-B) and **5.3** mAP (Gm-L), demonstrating the effectiveness of our method. In addition, although Base+GT uses the name of unknown categories, it performs poorly in the Aerial, Game, Medical, and Surgery datasets. Our method does not rely on unknown class names and significantly outperforms Base+GT (e.g., Surgery: 2.4 (Base+GT-L) vs 25.6 (UMB-Wb-L)). When compared with Base-FS, which received the same

| Setting | Aquatic | | | Aerial | | | Game | | | Medical | | | Surgery | | | Overall | | |
|---|---|---|---|---|---|---|---|---|---|---|---|---|---|---|---|---|---|---|
| | $U_{AP}$ | $U_{RE}$ | Avg/Std | $U_{AP}$ | $U_{RE}$ | Avg/Std | $U_{AP}$ | $U_{RE}$ | Avg/Std | $U_{AP}$ | $U_{RE}$ | Avg/Std | $U_{AP}$ | $U_{RE}$ | Avg/Std | $U_{AP}$ | $U_{RE}$ | Avg/Std |
| Mean Embedding+OOD | 4.2 | 76.8 | - | 4.8 | 16.9 | - | 22.4 | 80.6 | - | 0.3 | 2.8 | - | 17.0 | 95.5 | - | 9.7 | 54.5 | - |
| +ID Probability | 17.7 | 93.3 | - | 3.8 | 61.0 | - | 32.2 | 90.9 | - | 7.4 | 47.4 | - | 16.3 | 96.3 | - | 15.5 | 77.8 | - |
| +Gaussian (UMB-Gm) | 18.6 | 93.3 | 93.3/0 | 11.2 | 40.2 | 55.6/6.5 | 35.1 | 90.7 | 90.8/0.2 | 13.2 | 47.4 | 42.8/9.2 | 24.5 | 96.3 | 96.3/0 | 20.5 | 73.4 | 75.6/3.2 |
| /Weibull (UMB-Wb) | 18.6 | 93.3 | 93.3/0.1 | 11.1 | 41.3 | 55.9/6.1 | 32.7 | 90.7 | 90.8/0.2 | 8.6 | 47.4 | 42.2/10.2 | 25.6 | 96.3 | 96.3/0 | 19.3 | 73.8 | 75.7/3.3 |

Table 2: Ablation study of UMB on RWD. We provide incremental results of model performance. $U_{AP}$ and $U_{RE}$ represent the model's mean Average Precision (mAP) and corresponding recall rate on unknown category. Avg and Std denote the mean and variance of the recall rate distribution for unknown classes under different $\alpha$ settings (eqn. 15). In A.5, we provide more analysis.

supervision, FOMO did not achieve comprehensive leadership and even lagged behind by 3.6 mAP in Aquatic. Our method leads whether compared with Base-FS or FOMO, and in the Surgery dataset, we **doubled** the performance of FOMO (12.0 (FOMO-L) vs 25.6 (UMB-Wb-L)). Gm and Wb exhibit different strengths in various OWL-ViT models. UMB-Wb shows an advantage on the B/16 model (+1.1 mAP), while the trend is reversed on the L/14 base (-1.2 mAP). Therefore, we provide two different types of probability distributions (Gm and Wb) as interchangeable options.

### 4.4 Ablation Study

Table 2 provides the incremental results of our UMB. The initial performance uses the average category embedding (eqn. 10) and out-of-distribution probability (eqn. 14). When the in-distribution probability is introduced, which is used to capture the known attributes of the current target, the performance improves by 5.8 mAP. However, in the Aerial dataset, $U_{AP}$ only reaches 3.8 mAP, reducing by 1 mAP, replaced by a significant increase in recall rate (+44.1), which means that the detector erroneously treats many background samples as unknown objects. Such a result also proves the limitations of the unknown recall as the detection metric previously used in the OWOD benchmark. Finally, by adding the empirical distribution, our UMB achieves a comprehensive lead (Gm +5.0, Wb +3.8). In addition, the effect of balance parameter $\alpha$ on the recall rate is not obvious. In fact, in the Aquatic and Surgery datasets, the variance of the recall rate distribution reaches 0, which means that alpha correctly suppresses the background samples erroneously predicted by the detector. Overall, our UMB can provide a higher unknown recall rate (UMB-Gm 73.4, UMB-Wb 73.8) while ensuring detection accuracy.

### 4.5 Visualization

In figure 4, we provide the qualitative analysis of FOMO and our UMB. The visual analysis is divided into three parts: the recall ability of the unknown category, recall precision, and analysis of additional information. UMB shows superior performance in recalling unknown objects. UMB successfully recalled tools in the Surgery dataset (fifth row, Wb ID 3) and accurately recalled playgrounds and roofs in the Aerial dataset (second row, Wb ID 1, 2), while FOMO failed to recall these objects. Regarding recall precision, FOMO predicts multiple results to an object and incorrectly classifies unknown classified objects as known classes, such as the hero characters in the Game dataset (third row) and the misclassification of four objects in the Aquatic dataset (first row). In contrast, our UMB shows higher precision. Regarding additional information, the misclassification of FOMO in the Aquatic dataset reflects the defects of OWL-ViT in classifying these objects. With the help of formula n, UMB can infer the category most similar to the current object, and these categories correspond to the categories misclassified by FOMO. In addition, for the same object (ID 3,4), UMB identifies that the attribute with the greatest impact on the entire decision is consistent. These results prove the accuracy of our method in inferring the connection between unknown and known and discovering the attributes that have the greatest impact on decision-making.

## 5 Conclusion

This paper attempts to understand the detector's behaviour in predicting unknown objects. To achieve this, we propose a novel detection framework, UMB, which employs class-agnostic textual attributes to unearth potential objects in the background. Given that the model's detection process for known

and unknown classes hinges on textual attributes, our UMB can use the textual attributes of unknown objects to infer the most similar known category. In addition, we can calculate the attributes that have the most significant impact on the entire decision-making process. This supplementary information aids annotators in understanding the model's behaviour in predicting unknowns. We hope that UMB can promote the application of Open-World Object Detection in real-world scenarios.

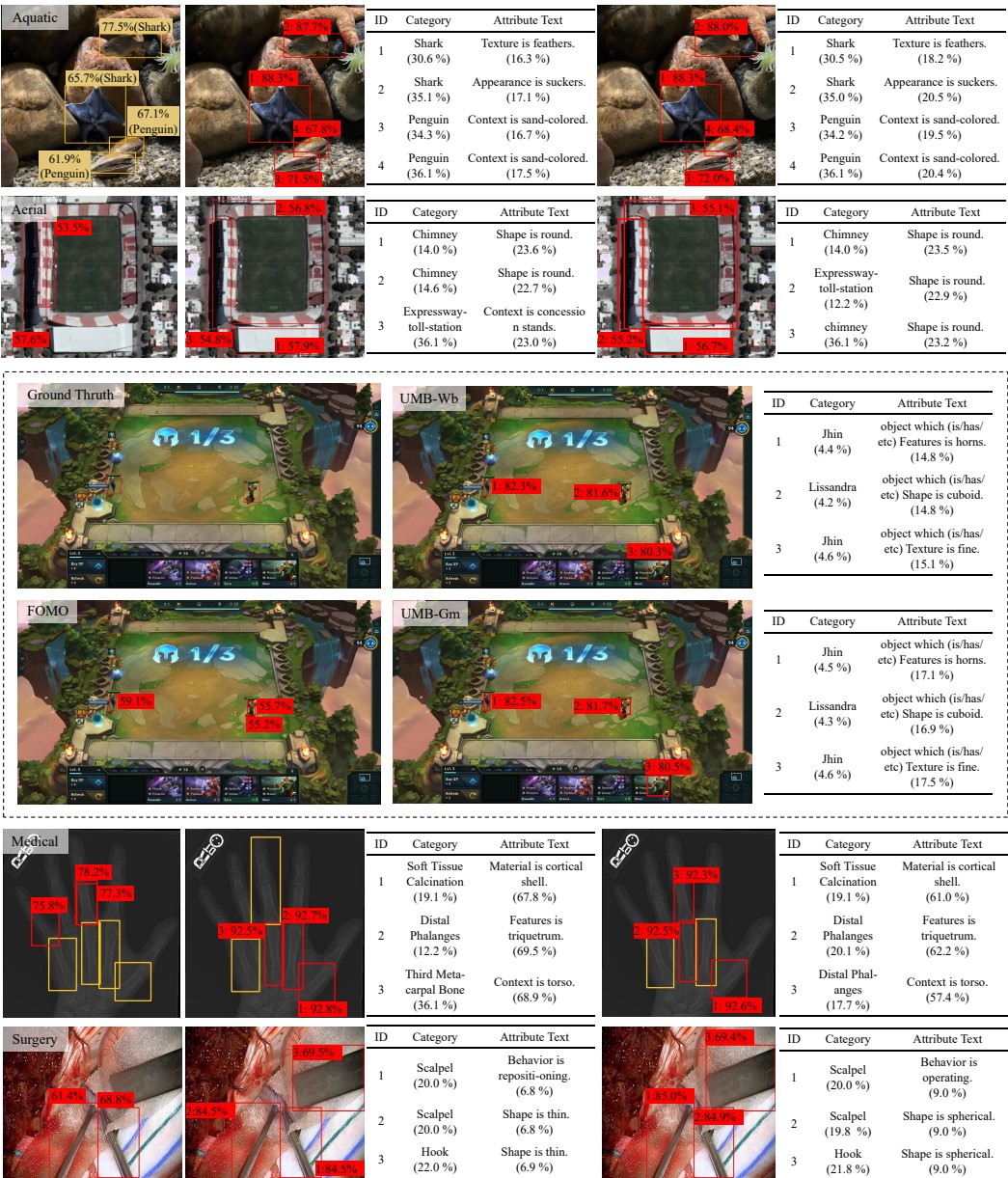

Figure 4: Qualitative Analysis. Each row, from left to right, represents: FOMO, UMB-Wb, and UMB-Gm, respectively. From top to bottom, the results are given for Aquatic, Aerial, Game, Medical, and Surgery. For fairness and clarity, we only display the TOP-K unknown predictions with a confidence level greater than 0.5. Unknown predictions are marked in Red, while known classes are marked in yellow. Each table provides the most similar known category (Category) for each unknown prediction, and the attribute (Attribute Text) that has the greatest impact on the decision-making process. In section 7, we provide an evaluation of the accuracy rate of similarity prediction.

## Acknowledgments

This work was supported in part by TCL Science and Technology Innovation Fund (Project No. 20231752).

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

# A    Appendix / supplemental material

**Algorithm 1:** Textual Attribute Generation and Known Class Prediction

/* Predefined attribute types */
$A$ = [Size, Shape, Behavior, ..., Appearance]
/* known class names (Aquatic) */
$C$ = [Jellyfish, Penguin, ..., Shark, Starfish]
$Attributes$ = []
/* attribute generation guided by LLM */
**for** $c_i$ *in* $C$ **do**
    **for** $a_j$ *in* $A$ **do**
        /* fill in the template (eqn. 1) */
        /* e.g. Penguin Size,
        Template(Penguin, Size) = I am using a language-vision model to identify {Penguin}. List the {Size} attributes of {Penguin}, which will be used for detection. */
        $Template(c_i, a_j) \to LLM$
        /* generate attributes, e.g, blue */
        $LLM \to attribute$
        /* collect attributes and predictions corresponding to LLM*/
        $(a_j, attribute) \to Attributes$
    **end**
**end**
/* encoding textual attribute to embedding */
$E_{att}$ = []
**for** $a_i, attribute_i$ *in* $Attributes$ **do**
    /* fill in the template (eqn. 2) */
    /* e.g. blue, Prompt(color, blue) = object which (is/has/etc) <color> is <blue>*/
    $Prompt(a_i, attribute_i) \to$
    $Text\_encoder \to E_{att}$
**end**
/* encoding image to visual embedding */
$E_{vis}$ = []
**for** $patch\_img$ *in* $image$ **do**
    $Vision\_encoder(patch\_img) \to E_{vis}$
**end**
/* attribute similarity */
$Sims$ = []
**for** $e_{vis_k}$ *in* $E_{vis}$ **do**
    $sims_i$ = []
    **for** $e_{att_i}$ *in* $E_{att}$ **do**
        /* sim denotes the cosine similarity */
        $sim(e_{vis_k}, e_{att_i}) \to sims_i$
    **end**
    $sims_i \to Sims$
**end**
/* known class prediction */
$output$ = []
**for** $idx, sims_i$ *in* $Sims$ **do**
    /* trained attribute weight $W$*/
    $W[idx] \cdot sims_i \underset{eqn.3}{\to} output$
**end**
**return** $output$

**Algorithm 2:** Text Attribute Modeling (TAM)

$D_{img} = \{image_1, ..., image_m\}$
$Known\_classes = \{C_1, ..., C_{kn}\}$
$f = \{C_1: \{e_{att_1}: [], ..., e_{att_n}: []\},$
    $..., C_{kn}: \{...\}, C_u: \{...\}\}$
**for** $image_i$ *in* $D_{img}$ **do**
    Vision_encoder$(image_i) \to E_{vis}$
    **for** $e_{vis_k}$ *in* $E_{vis}$ **do**
        **for** $e_{att_i}$ *in* $E_{att}$ **do**
            **for** $C_j$ *in* $Known\_classes$ **do**
            /* building mapping: $f_{j,i}$ */
            $(\underbrace{sim(e_{vis_k}, e_{att_i})}_{Cosine\ Similarity}, \underbrace{p(C_i|e_{vis_k})}_{eqn.3})$
            $\underset{eqn.4}{\to} f[C_j][e_{att_i}]$
            **end**
        **end**
    **end**
**end**
/*eqn. 5*/
**for** $x$ *in* $Range(-1, 1, gap)$ **do**
    **for** $e_{att_i}$ *in* $E_{att}$ **do**
        $Max\_val = 0$ **for** $C_j$ *in* $Known\_classes$ **do**
            **for** $val_i$ *in* $f[C_j][e_{att_i}]$ **do**
                **if** $val_i$ *in* $[x, x + gap]$ **then**
                    $max(val_i, Max\_val) \to$
                    $Max\_val$
                **end**
            **end**
        **end**
        $(x, Max\_val) \to f[C_u][e_{att_i}]$
    **end**
**end**
/* eqn. 6, here we set gap to 0.0001 */
**for** $x$ *in* $Range(-1, 1, gap)$ **do**
    **for** $e_{att_i}$ *in* $E_{att}$ **do**
        **if** $(x, 0)$ *in* $f[C_u][e_{att_i}]$ **then**
            $f[C_u][e_{att_i}]$ $del$ $(x, 0)$
            $Linear\ Interpolation \to$
            $(x, estimate) \to f[C_u][e_{att_i}]$
        **end**
    **end**
**end**
/* eqn. 7 */
**for** $x$ *in* $Range(-1, 1, gap)$ **do**
    **for** $e_{att_i}$ *in* $E_{att}$ **do**
        $f[C_u][e_{att_i}] \underset{filter}{\leftarrow} Sliding\ Window$
    **end**
**end**
**for** $e_{att_i}$ *in* $E_{att}$ **do**
    $PMM_i \underset{training}{\leftarrow} (f[C_u][e_{att_i}])$
**end**
**return** $[PMM_1, PMM_2, ..., PMM_n]$

In this section, we supplement the details omitted in the main text.

## A.1 Task Formulation

In the context of OWOD, the detection task is divided into a series of subtasks $T = \{T_1, T_2, ..., T_{|T|}\}$ and their corresponding categories $K = \{K_1, K_2, ..., K_{|T|}\}$. $T_i$ includes all known categories from previous tasks and introduces new categories on this basis: $K_i = (\bigcup_{j=1}^{i-1} K_j) \cup K_{new}$, where $K_{new}$ denotes introduced new categories. When the model is trained on $T_i$, our expectation is that the model should be able to detect all categories it has encountered so far (i.e., $K_i$), as well as discover those unlabelled but interesting categories. For the purpose of evaluation, the interest object is defined as those that belong to $K$ but not to $K_i$ (i.e., $K - K_i$).

## A.2 Datasets

The OWOD benchmark is a combination of COCO[30] and VOC[31] datasets. In the era of foundation models, the zero-shot capabilities of detectors on OWOD benchmark have even reached their limits. Therefore, consistent with FOMO[27], we will switch the evaluation benchmark to RWD. The RWD benchmark consists of five typical application scenarios for object detection, including underwater scenes, representing visual blurring caused by the environment (Aquatic[32]); aerial scenes, where the targets are small and difficult to distinguish (Aerial[33]); scenarios using synthetic data when data is lacking (Game[34]); medical X-ray scenes, where it is difficult to distinguish between categories and professional knowledge is required(Medical[35]); and human surgery scenes, where the field of view is blurred by blood (Surgery[36]). The detailed division of the RWD benchmark is shown in Table 3. We divide RWD into two subtasks according to a 50% category ratio. When training in Task 1, all categories in the test set that belong to Task 2 are treated as unknown classes, and when training in Task 2, the categories of Task 1 are considered as previously seen classes.

## A.3 Metric

For known categories, we adopt the widely used mean Average Precision (mAP) as the evaluation metric for object detection. For unknown classes, previous OWOD methods[5, 4] used the recall rate of unknown classes as the evaluation metric. However, such a metric leads to models greedily treating all background objects as potential samples. Therefore, we adopt mAP, consistent with the evaluation metric for known classes, which simultaneously assesses the detector's recall ability for unknown classes and the precision of the predictions.

## A.4 Details

All experiments were conducted using a single NVIDIA GeForce RTX 4090 GPU. Following FOMO, we initialized with the frozen OWL-ViT[19] (L/14 and B/16), which was trained on a mixed dataset composed of Object 365[37] and Visual Genome[38], demonstrating strong generalization capabilities. The large language model used for attribute generation is GPT-3.5. These attributes were matched with all predictions in the dataset, and the corresponding visual embeddings were collected if the IOU exceeded the threshold (0.8). The average of these visual embeddings was calculated to obtain the average embedding of the attributes. Following FOMO, we adopted attribute selection, attribute adaptation, and attribute refinement to train the linear combination weight.

All optimizers used AdamW. During the attribute selection phase, BCE was the loss function, and the learning rate remained constant without decreasing with iterations. The attribute selection phase reduced the number of attributes. Based on the ranking of weights after training, only the top 25 attributes per attribute type were retained. Attribute adaptation was used to narrow the distance between the text attributes and the average embedding of the dataset. This phase used MSE as the loss function, with a maximum of 1000 iterations. Attribute refinement took the text embedding as the parameter to be optimized, with BCE as the loss function. Attribute refinement narrowed the distance between the text embedding and the visual embedding. During the attribute selection and attribute refinement phases, the learning rate and maximum number of iterations for training were set to three values ([1e-5, 5e-5, 1e-4], [1, 10, 100]), iterating over these settings during each training to select the optimal setting.

| Dataset | Task1 known And Task2 Previous known | Task2 known And Task1 unknown |
|---|---|---|
| Aquatic | Fish(100, 100, 1372), Jellyfish(93, 93, 398) Shark(100, 100, 179), Penguin(100, 100, 306) | Puffin(100, 172), Stingray(68, 85), Starfish(43, 57) |
| Aerial | Vehicle(100, 100, 8350), Storagetank(100, 100, 6229) Stadium(100, 100, 277), Ship(100, 100, 12420), Groundtrackfield(100, 100, 641), Golffield(100, 100, 222) Dam(100, 100, 225), Basketballcourt(100, 100, 617) Airport(100, 100, 287), Airplane(100, 100, 2423) | Expressway-Service-area(100, 464) Expressway-toll-station(100, 302) Baseballfield(100, 1192), Windmill(100, 1078) Bridge(100, 809), Chimney(100, 334) Harbor(100, 1072), Overpass(100, 684) Tenniscourt(100, 2583), Trainstation(100, 207) |
| Game | Gankplank(31, 31, 30), Poppy(21, 21, 32) Blitzcrank(28, 28, 28), Illaoi(24, 24, 23) Singed(35, 35, 37), Zac(25, 25, 27) Janna(39, 39, 33), Ezreal(38, 38, 32) Twitch(25, 25, 25), Camille(29, 29, 17) Twisted Fate(18, 18, 31), Jayce(29, 29, 24) Swain(33, 33, 24), Caitlyn(22, 22, 24) Lulu(21, 21, 28), Trundle(25, 25, 33) Warwick(29, 29, 28), Zilean(30, 30, 25) Katarina(25, 25, 26), Vex(23, 23, 32) Ziggs(29, 29, 29), Braum(26, 26, 25) Darius(16, 16, 37), Cho-Gath(22, 22, 29) Tristana(28, 28, 36), Kassadin(22, 22, 23) Malzahar(23, 23, 24), Heimerdinger(26, 26, 30) Vi(32, 32, 37), Veigar(21, 21, 23) | Talon(17, 22), Lux(23, 18, Seraphine(19, 17), Jhin(15, 19) Taric(16, 22), Leona(23, 19) Viktor(15, 18), Lissandra(17, 25) Yuumi(18, 28), Akali(17, 17) Ekko(17, 21), Samira(19, 20) Kai-Sa(24, 23), Dr- Mundo(18, 23) Fiora(14, 20), Orianna(19, 22) Jinx(16, 19), Yone(19, 20) Quinn(22, 18), Miss Fortune(23, 21) Sion(22, 15), Kog-Maw(23, 22) Garen(21, 20), Graves(17, 19) Urgot(23, 24), Galio(24, 18) Shaco(14, 28), Zyra(18, 20) Tahm Kench(23, 14) |
| Surgery | BipolarForcepsUpSkeleton(100, 100, 361) SuctionTubeSkeleton(100, 100, 812) Retractors(100, 100, 259) BipolarForcepsUp(100, 100, 508) BipolarForcepsDown(100, 100, 496) SuctionTube(100, 100, 872) | Curette(59, 57), Hook(100, 123) PliersDown(92, 94), Scissors(38, 30) Scalpel(95, 90), PliersUp(98, 98) BipolarForcepsDOwn(1, 0) |
| Medical | Second metacarpal bone(86, 86, 96) Fifth metacarpal bone(82, 82, 95) Distal phalanges(100, 100, 481) Third metacarpal bone(80, 80, 95) Proximal phalanges(100, 100, 475) Intermediate phalanges(96, 96, 376) | Soft tissue calcination(38, 50), Ulna(78, 90) Fourth metacarpal bone(83, 95), Artefact(2, 3) First metacarpal bone(80, 94), Radius(76, 92) |

Table 3: Detailed explanation of the dataset split. Each dataset is split into two subtasks, each maintaining a category proportion of 50%. When training on Task 1, the categories of Task 2 are treated as unknown classes. During training in Task 2, the classes from Task 1 are labeled as previously seen categories, and new classes divided into Task 2 are introduced. The numbers (num1, num2, num3) following each category in Task 1 represent the number of training instances in Task 1, Task 2, and the test set, respectively. The numbers (num1, num2) in Task 2 represent the number of training and test instances for this category in Task 2.

Upon completion of training, the detector could detect known classes. To detect unknowns, we established the empirical probability for each attribute. In the distribution optimization phase, we set the window value to 10. In the distribution fitting phase, we used two different probability models (Gaussian and Weibull). During training, Adam was used as the optimizer, the learning rate was set to 0.01, the maximum number of iterations was 10000, and the maximum number of probability models was set to 5. Since the purpose of fitting was to capture the shape of the empirical probability distribution and establish a continuous probability distribution, we set an interval of 0.0001 between -1 and 1. This setting was used to sparsify the data of the original distribution. Then, distribution optimization and fitting were performed in the sparsified distribution.

## A.5   Comprehensive Ablation Experiments

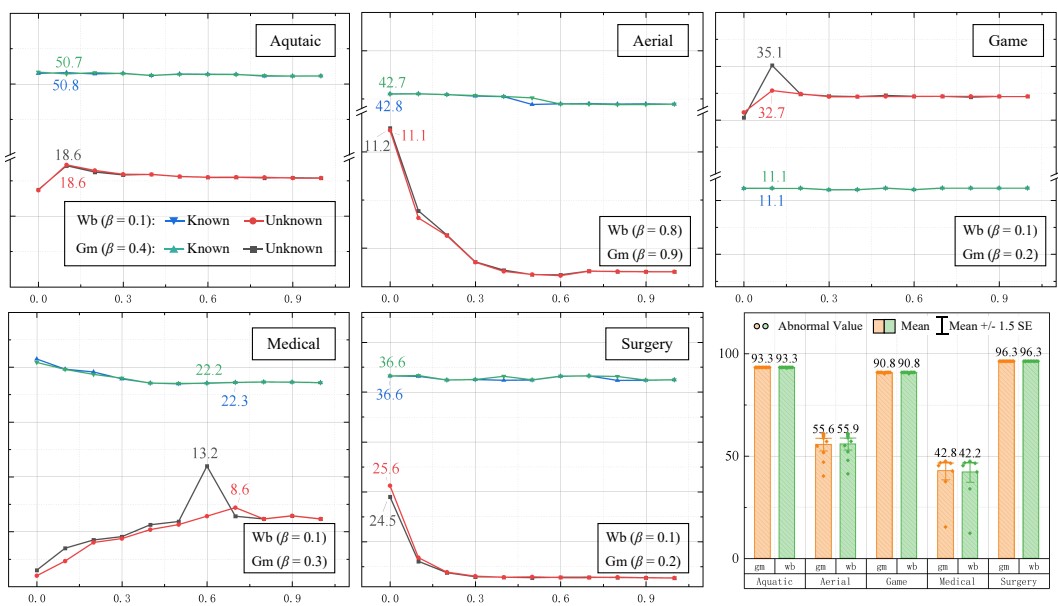

Figure 5: Subfigures 1 through 5 present the performance variations of the model under different settings of $\alpha$ (eqn. 15). For clarity, we only display the performance changes corresponding to the optimal $\beta$ (eqn. 4). Subfigure 6 provides the statistics of the recall rates corresponding to these five figures as $\alpha$ varies. Here, Mean represents the average, while SE denotes the standard deviation.

Figure 5 illustrates the model's performance under different $\alpha$ settings with the optimal $\beta$. In addition, subfigure 6 provides the statistics of the recall rates corresponding to subfigures 1 through 5. In the Aquatic and Game datasets, the performance of UMB is not sensitive to the changes in $\alpha$, showing minor performance differences. This is because Aquatic is consistent with the training data of OWL-ViT, and Game is synthetic data, neither of which poses additional challenges (such as target size). However, in the remaining datasets, the performance of UMB shows significant changes. For example, the performance of the Aerial dataset drops from 11.1 mAP (wb) to 3.6 mAP. These datasets have significant differences from the OWL-ViT training dataset, and their environmental characteristics (such as small objects in Aerial, similar objects in Medical, and blood-contaminated backgrounds in Surgery) pose additional challenges to the model. These constraints make UMB sensitive to changes in $\alpha$. Nevertheless, a reasonable $\alpha$ can balance in-distribution and empirical probability contributions to achieve better detection performance.

For the recall rate of unknown categories, except for the Aerial and Medical datasets, UMB maintains a high value (93.3% in Aquatic) and remains stable in the remaining datasets. This indicates that our method does not predict more objects under different $\alpha$ settings but predicts potential objects more accurately. In the Aerial and Medical datasets, the model's recall rate is almost half that of other datasets, and these recall rates show significant fluctuations with the change of $\alpha$. Therefore, we infer that when the model's recall rate for a specific dataset is high, the balance between intra-division

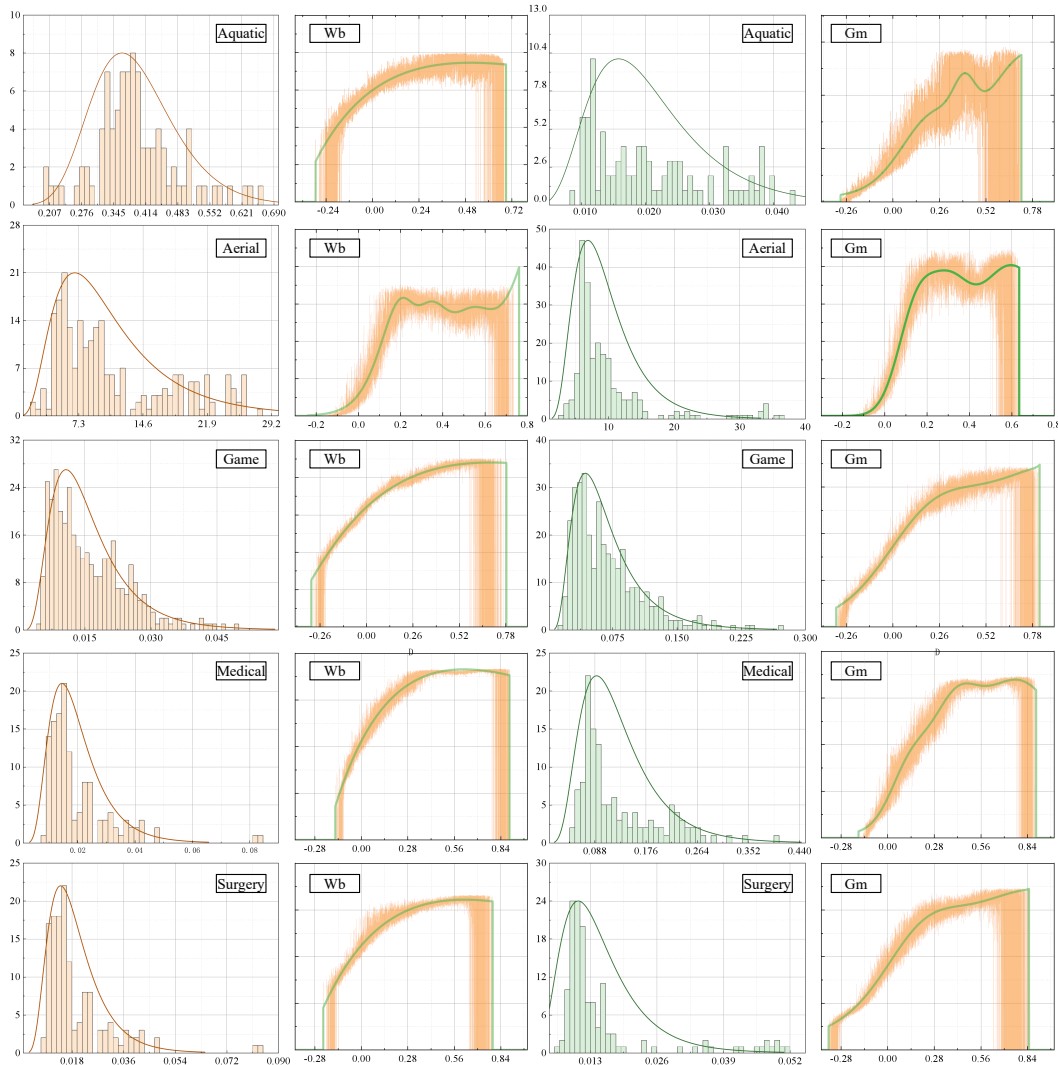

Figure 6: Training result for the Probabilistic Mixture Model. The training results for Aquatic, Aerial, Game, Medical, and Surgery are represented from top to bottom. We only display the fitting process corresponding to the optimal $\beta$ setting for clarity. The settings for $\beta$ can be found in figure 5. Each original distribution is first filtered for noise using distribution optimization (linear interpolation and sliding window) and then fitted with the basic probability model (Weibull or Gaussian) to the optimized distribution. The first and third images in each row show the MSE loss distribution corresponding to convergence, while the second and fourth images present the best-fitting results across all attributes. In the best-fitting result, the yellow line represents the original distribution, and the green line represents the result after fitting the probability model.

probability and empirical probability has a relatively small impact on the recall rate; when the recall rate is low, this balance has a more significant impact on the recall rate.

## A.6 Probabilistic Mixture Model

Figure 6 presents the fitting results of the probabilistic mixture model. When the original probability distribution exhibits multiple peaks, the MSE loss of the probability fitting may stabilize at a relatively high value. For instance, in the first column of the second row, the original probability distribution is composed of more than three elemental probability distributions. Upon completion of training, the MSE value stabilizes around 7.3, indicating that the model faces challenges when fitting distributions

with multiple peaks. However, when the original probability distribution exhibits a single peak, the MSE loss approaches zero, demonstrating the model's advantage in fitting unimodal distributions. For example, in the Game dataset, the MSE value of the Gaussian model stabilizes around 0.015. Despite the possibility of multiple peaks in the original distribution, both the Weibull and Gaussian distribution can capture its essential shape characteristics. For instance, in the Aerial dataset, the Gaussian model can fit two smooth peaks to represent the original distribution. Similarly, the Weibull model can fit a smooth curve when facing distributions with multiple peak features. However, in the case of unimodal distribution, both the Gaussian and Weibull models can capture the original distribution's unimodal feature and fit a smooth curve with a single peak. For example, in the Surgery dataset, the Weibull and Gaussian models in the Game datasets demonstrate this ability.

## A.7 Similarity Evaluation

In order to understand the behavior of the model, we provide the known class most similar to the unknown object and the attribute with the most significant impact on decision-making. Through visualization, we observe that many known predictions overlap with unknown objects in the detector's predictions. These known predictions represent that the detector considers the current object to belong to a known category. Therefore, we estimate the model's accuracy in similar inference by

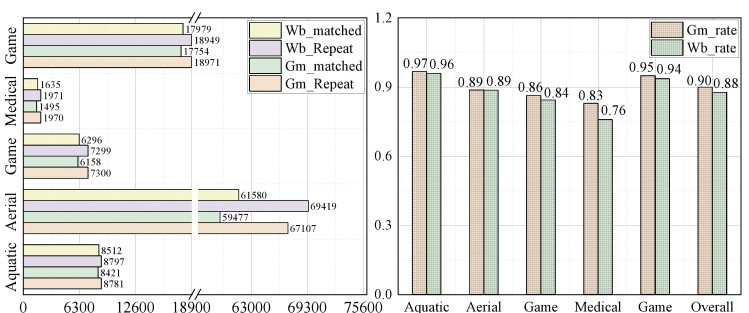

Figure 7: Evaluation of similarity. The left figure represents the number of predictions where unknown and known categories overlap. The right figure shows the accuracy of the model's inference among these numbers.

whether the inferred known class is consistent with these overlapping known predictions. Specifically, if their IoU exceeds 0.95, we consider these two predictions overlapping. Then, if the known class inferred by the detector is consistent with the known prediction, we consider the model inference correct. Otherwise, it indicates an inference error. As shown in Figure 7 (left), the detector has many duplicate predictions for unknown and known classes. Among these duplicate predictions, the proportion of correctly inferred predictions is large, more than 80% (Figure 7). Compared with Wb, Gm's accuracy rate is always lower. In the Aquatic dataset, Gm lags behind Wb by 0.1 percentage points in accuracy, and in Medical, it lags by 0.6 percentage points. Overall, both Gm and Wb show high inference accuracy (about 90%), and compared with Gm, Wb shows a higher accuracy rate (about 2%).

## A.8 Attribute Study

Figure 8 and Figure 9 present the results of attribute analysis. For clarity, we have selected only the top three categories with the highest prediction counts from the detector and the top five attributes that have the most significant impact on them. In each row, the top part shows the prediction counts for these categories. For instance, Puffin (252) in Aquatic indicates that the detector gave 252 predictions for the Puffin category. The left side shows the number of times these attributes have been identified as having the most significant impact. Moreover, the right side represents the attributes with the highest average inference scores from the model.

UMB demonstrates a strong ability to capture object attributes. In the Surgery dataset, the attribute (Behavior is repositioning) dominated UMB-Wb's 127 predictions (total 274) for the Hook category, and this number rose to 134 (total 273) in UMB-Gm, accounting for nearly half. This implies that when an object exhibits such an attribute, the detector will likely predict it as the Hook category. UMB exhibits precise attribute discrimination capabilities. Since the right side of each row presents the average score of the attribute's impact on all detector predictions, their differences are insignificant. Nevertheless, in the Aquatic dataset, UMB-Gm still distinguished these attributes. For instance, the

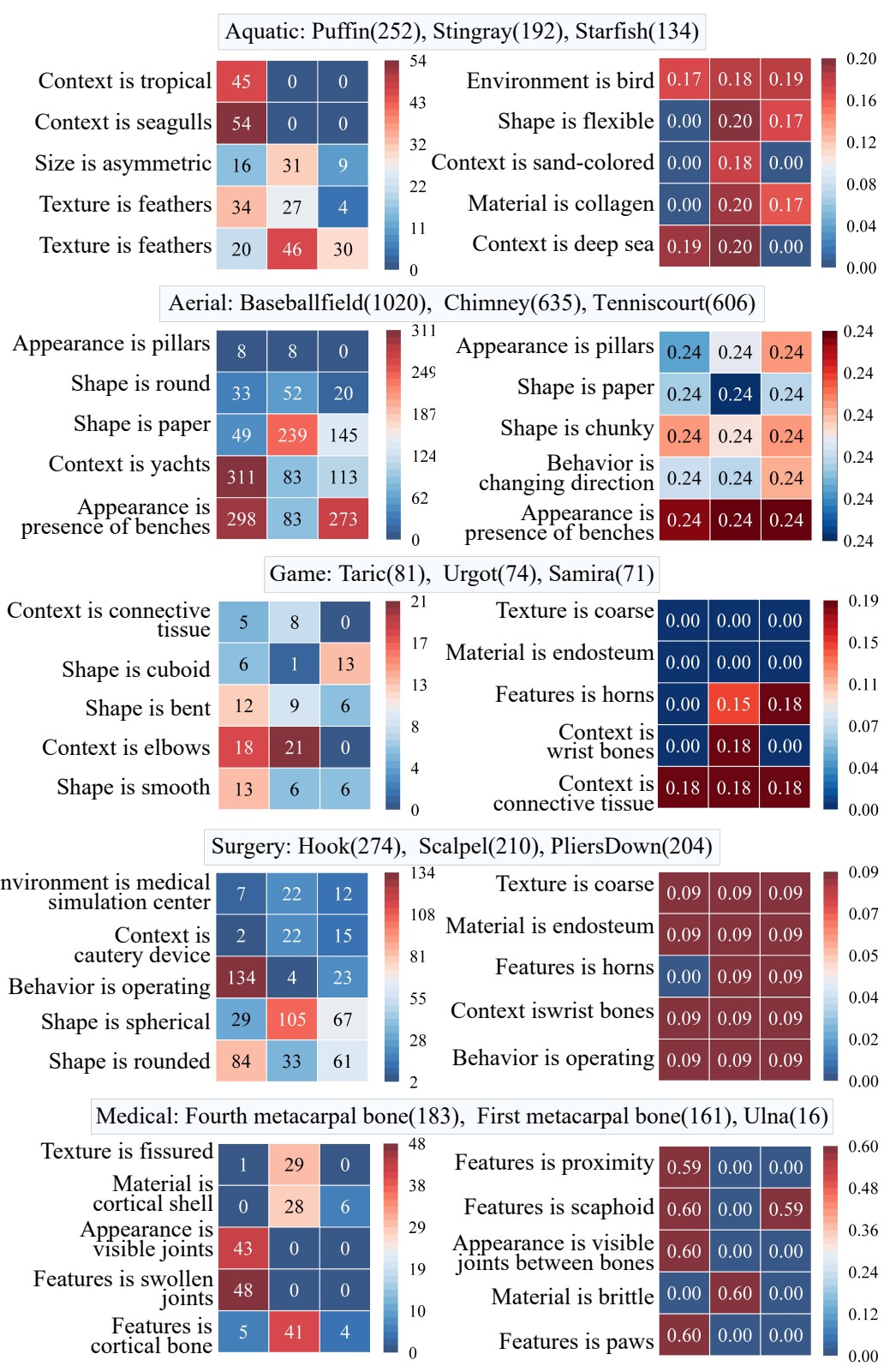

Figure 8: Cross-category attribute analysis (Gm). Each row from top to bottom presents the results for Aquatic, Aerial, Game, Surgery, and Medical. The left in each row represents the number of times the attribute influences the decision, while the right side indicates the average score of the attribute.

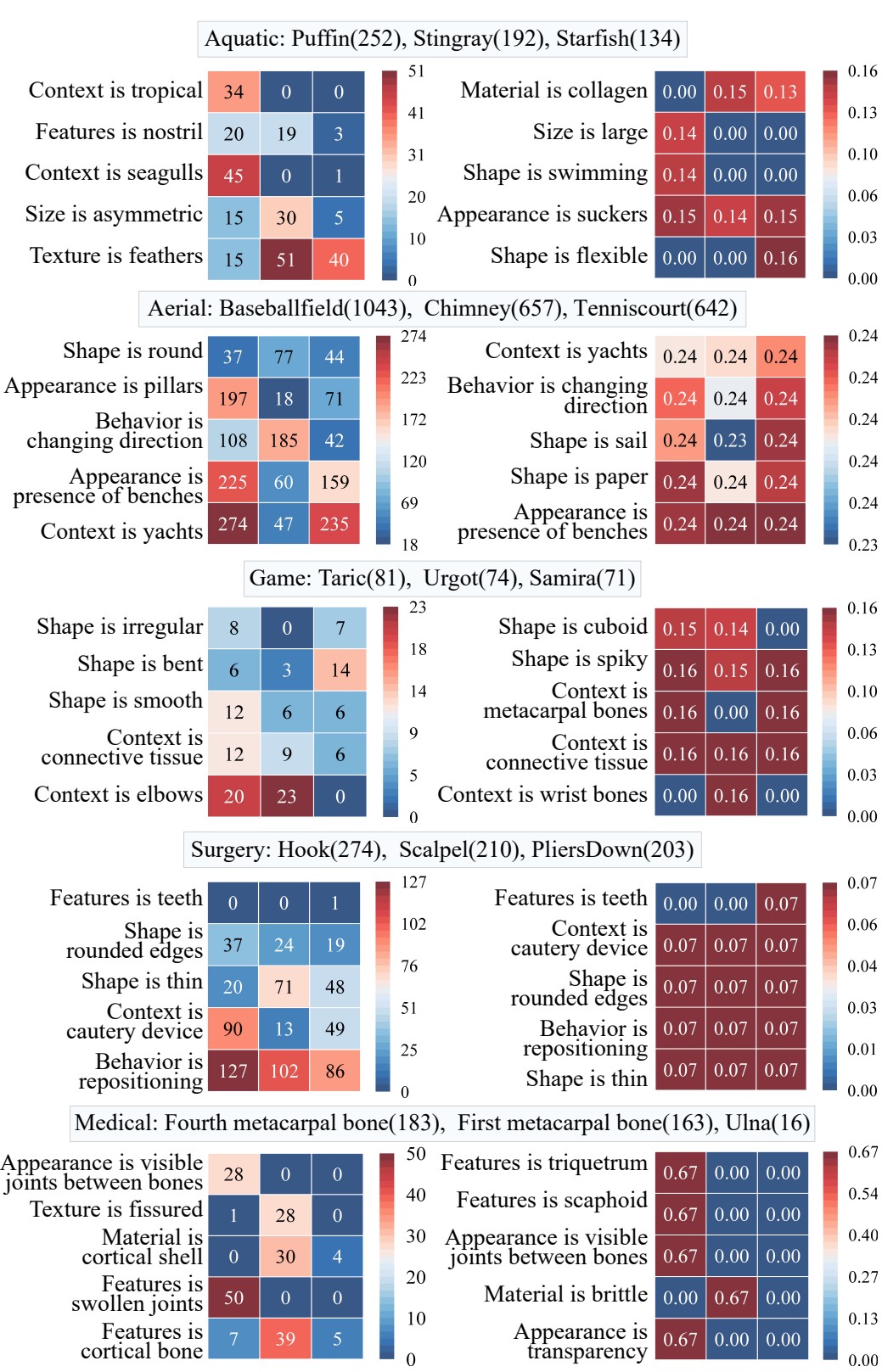

Figure 9: Cross-category attribute analysis (Wb). Each row from top to bottom presents the results for Aquatic, Aerial, Game, Surgery, and Medical. The left in each row represents the number of times the attribute influences the decision, while the right side indicates the average score of the attribute.

attribute (Context is the deep sea) had an average score of 0.2 in Puff but dropped to 0 in Starfish. A similar scenario occurred in UMB-Wb, where the attribute (Shape is flexible) had an impact of 0.16 on Starfish, but it dropped to 0 for Puffin. Overall, both UMB-Gm and UMB-Wb can capture and distinguish object attributes and make corresponding predictions based on these attributes.

We conducted additional comparative experiments on Linear Interpolation (LI) and Sliding Window (SW), as detailed in Table 4 of the attached document. The results indicate that employing LI or SW independently does not lead to significant performance improvements. Both methods exhibit only marginal enhancements compared to the original approach, suggesting that LI and SW, when used in isolation, are insufficient for accurately modeling the data distribution.

| Setting | Aquatic | Aerial | Game | Medical | Surgery |
|---------|---------|--------|------|---------|---------|
| ME+OOD  | 4.2     | 4.8    | 22.4 | 0.3     | 17.0    |
| +ID     | 17.7    | 3.8    | 32.2 | 7.4     | 16.3    |
| PMM(OD) | 18.0    | 4.1    | 32.3 | 7.5     | 18.3    |
| PMM(LI) | 18.0    | 4.1    | 32.3 | 7.7     | 18.4    |
| PMM(SW) | 17.8    | 5.0    | 32.3 | 7.1     | 18.4    |
| PMM(Gm) | 18.6    | 11.2   | 35.1 | 13.2    | 24.5    |
| PMM(Wb) | 18.6    | 11.1   | 32.7 | 8.6     | 25.6    |

Table 4: Ablation Studies. ME is Mean Embedding, OOD is Out-of-Distribution. PMM(OD) uses original distribution, LI, SW use Linear Interpolation, Sliding Window. Gm, Wb are filtered models, denoting Gaussian, Weibull distribution.

## A.9   Broad Impact

In this paper, we focus on the performance of detectors in open-world object detection and attempt to understand the model's behavior when predicting unknown categories. Our approach can help annotators gain a deep understanding of the model's decision-making process, thereby guiding subsequent optimization work and improving the overall performance of the detector. At the same time, understanding the model's behavior may expose potential flaws malicious actors could exploit for illegal activities. For this reason, we choose to open-source our code, both to promote the development of the current field and to identify and prevent these potential issues through the power of the community.

## A.10   Limitations

Our focus is on understanding the behaviour of model predictions. Hence, we attempt to migrate the OVC detector to the OWOD task. As our method does not directly train the weights of the OVC detector but merely processes its output, the performance ceiling of our method will be constrained by the inherent performance of the OVC detector itself. Furthermore, due to the visual-text alignment relationships of the OVC requiring extensive data training, fine-tuning on actual application datasets could lead to additional annotation costs.

## A.11   Failure Cases

We present typical cases of detection failures from each dataset, focusing on recall capability and detection accuracy. For instance, in the Aquatic dataset, UMB failed to detect small orange fish, while in the Aerial dataset, it did not successfully recall vehicles. These instances reveal the detector's shortcomings in recall capability. Moreover, in the Game and Surgery datasets, UMB displayed occurrences of repeated predictions. Nevertheless, UMB still outperformed FOMO in overall performance. Specifically, in the Aquatic dataset, UMB accurately located the contours of the fish, whereas FOMO showed deviations in contour localization and even missed similar objects. Furthermore, FOMO incorrectly identified the reflection of the photographer's shoes in the glass as an unknown object, demonstrating lower precision. Similar issues were observed in the Aerial and Game datasets, where FOMO often confused objects with the background, resulting in new erroneous predictions, such as misidentifying rooftops as a single object in the Aerial dataset. However, UMB did not commit the same errors in these cases. In summary, although UMB also exhibited some false detections in certain scenarios, it outperformed FOMO in both detection accuracy and the ability to recall potential objects.

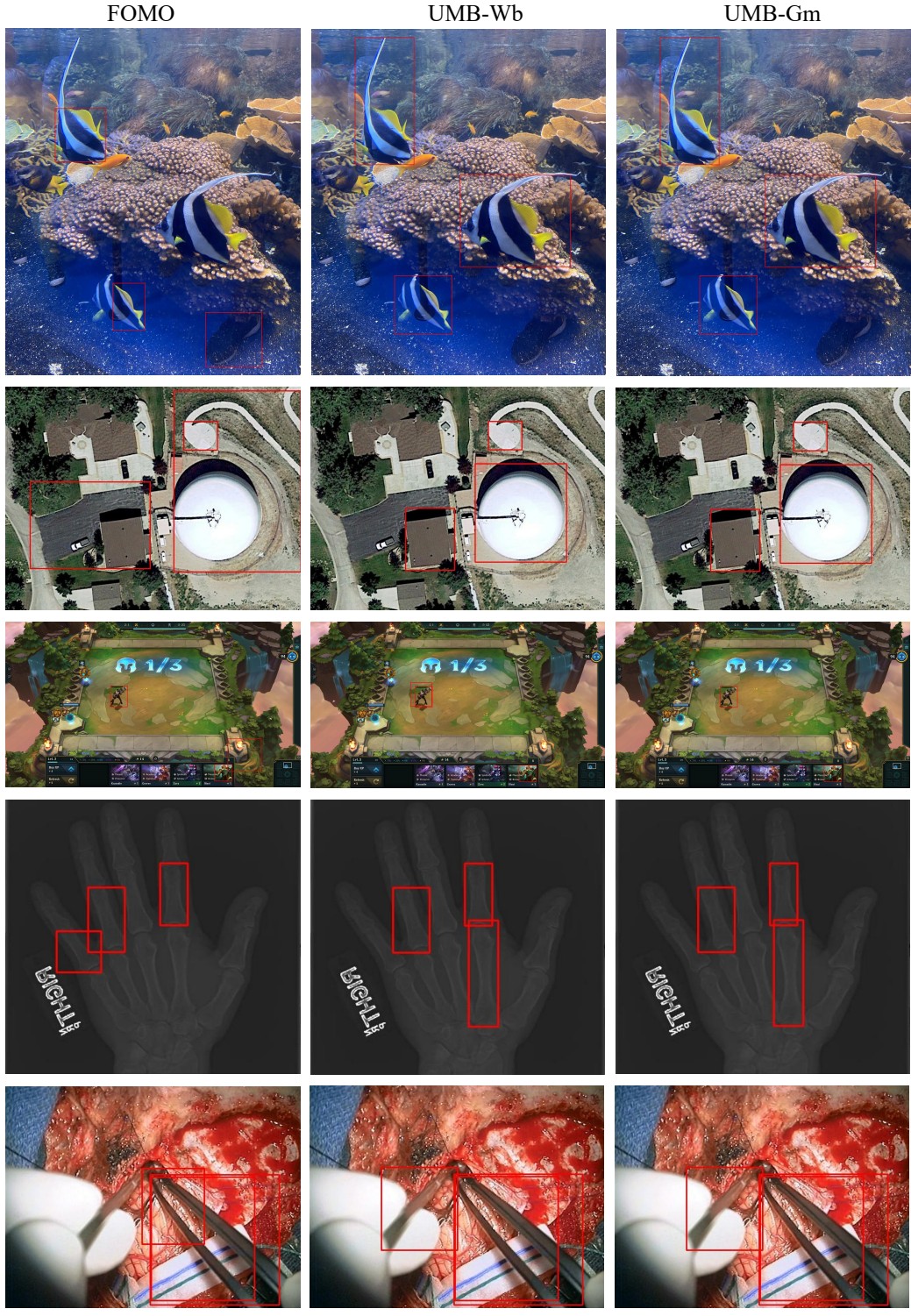

Figure 10: Qualitative Results (Failure Cases). From top to bottom, the datasets are Aquatic, Aerial, Game, Medical, and Surgery. To ensure clarity and fairness in comparison, we only display predictions for unknown, selecting those with a confidence score greater than 0.5 and ranked within the top K unknown category predictions. The results indicate that the UMB method demonstrates higher precision and recall in addressing the FOMO problem.

