# OpenReview forum: "UMB: Understanding Model Behavior for Open-World Object Detection"
_NeurIPS.cc/2024/Conference — NeurIPS 2024 poster_

### Official Review · Reviewer_sXkA · 2024-07-02

**Soundness:** 3
**Presentation:** 2
**Contribution:** 2
**Rating:** 6
**Confidence:** 3

**Summary:**

The paper presents Understanding Model Behavior (UMB), a framework for Open-World Object Detection (OWOD) that not only detects unknown objects but also analyzes the decision-making with text attributes. UMB first prompts large language models to generate text attributes. Then, it models the empirical, in-distribution, and out-of-distribution probabilities to estimate the likelihood of an object being predicted as a foreground. Based on this, UMB can infer the similarity of unknown objects to known classes, thereby identifying the most important attributes. It significantly improves over previous state-of-the-art methods on the Real-World Object Detection (RWD) benchmark, demonstrating its effectiveness in identifying unknown objects.

**Strengths:**

1. The motivation for modeling fine-grained text attributes in open-world learning is reasonable and interesting.
2. The performance gains on benchmarks are significant in both known and unknown classes.
3. The method can predict attributes for unknown classes, benefitting practical usage.

**Weaknesses:**

1. The compared methods are not convincing, lacking some more advanced open-world object detection works.
2. The ablative studies are not sufficient.
3. There are some typos leading to the reading difficulty.

**Questions:**

1. The performance comparison in Table 1 lacks some advanced open-world object detectors [1-3]

2. It is suggested that more ablative studies be conducted to justify fine-grained designs, including the linear interpolation in PMM, different texture attribute designs, and shift & scale in Eqn.11. The corresponding experimental analysis should also be given.

3. There are some mismatches between Fig. 2 and the method descriptions. I cannot find Class-agnostic Text Embedding in Sec 4.1, and I am confused about the Fig.3 caption "To establish To establish."

4. The literature review should cover more recently published works on open-set learning for object detection [1-4].

[1] Zohar, O., Wang, K. C., & Yeung, S. (2023). Prob: Probabilistic objectness for open-world object detection. In Proceedings of the IEEE/CVF Conference on Computer Vision and Pattern Recognition (pp. 11444-11453).

[2] Wang, Z., Li, Y., Chen, X., Lim, S. N., Torralba, A., Zhao, H., & Wang, S. (2023). Detecting everything in the open world: Towards universal object detection. In Proceedings of the IEEE/CVF Conference on Computer Vision and Pattern Recognition (pp. 11433-11443).

[3] Ma, S., Wang, Y., Wei, Y., Fan, J., Li, T. H., Liu, H., & Lv, F. (2023). Cat: Localization and identification cascade detection transformer for open-world object detection. In Proceedings of the IEEE/CVF Conference on Computer Vision and Pattern Recognition (pp. 19681-19690).

[4] Li, W., Guo, X., & Yuan, Y. (2023). Novel Scenes & Classes: Towards Adaptive Open-set Object Detection. In Proceedings of the IEEE/CVF International Conference on Computer Vision (pp. 15780-15790).

**Limitations:**

No potential negative societal impact.

---

> ### Author Rebuttal · Authors · 2024-08-03
>
> We sincerely appreciate the valuable feedback you provided on our work, particularly regarding the comparisons with recent works, the ablation experiments, and the content discrepancies. Below is our detailed response to each of your comments:
>
> **Comparison with Recent Works**:  The references [1]-[4] you mentioned include three distinct areas: open-set object detection [4], open-vocabulary object detection (OVC) [2], and open-world object detection (OWOD) [1, 3].  Open-set object detection (OSOD) requires the detector to be trained on a closed set without any annotations of unknown objects. During testing, the detector is required to detect both known and unknown objects. OVC (Section 2.1) narrows the distance between regions and text during training, leveraging pre-trained models on large-scale datasets and using pre-trained encoders (e.g., CLIP) to enhance generalization. In testing, the matching between customized text and regions determines the class to which a region belongs.  OWOD imposes stricter requirements, involving two components: detecting unknown objects of interest and incremental learning. The former, similar to OSOD, demands the detector to identify unknown objects during inference, while the latter requires the detector to classify some unknown objects as known classes and fine-tune the model to accommodate the new classes. These two processes simulate real-world applications where unknown objects are continuously labeled, and annotators select interesting targets for incremental learning, thereby enhancing the detector's performance. We have placed the problem definitions in Appendix (A.1), and a diagram illustrating the annotator's involvement in the entire process is provided in Figure 1.
>
> Among these references, those relevant to our work are [1, 3]. However, these methods do not utilize the foundation model, training from randomly initialized weights, resulting in poor performance. For example, in the COCO and VOC mixed M-OWODB, OWL-Vit achieves  79% unknown recall with the assistance of LLM, while PROB only achieves 19.4%. This result was obtained under the zero-shot setting of OWL-Vit, and if OWL-Vit were fine-tuned, the gap would further widen. Therefore, we follow FOMO and convert the evaluation dataset to the RWD benchmark. The RWD is more complex and adheres to the few-shot setting, where using detectors like PROB and CAT, which are trained from random weights, is less appropriate. In fact, when we simply implemented PROB on the RWD, we found it could not even detect unknown classes. Thus, we did not compare PROB and CAT because they rely on large amounts of training data, which is not compatible with the few-shot setting of RWD. We have discussed more relevant and recent methods, such as FOMO, and foundation models with the same supervision in the paper, with technical details provided in the appendix. If you have further questions, please refer to the appendix, where we explain the problem definition, benchmark setup, and additional experiments in more detail. Furthermore, we have cited PROB and CAT in our paper, as seen in references [20] and [5].
>
> **Ablation Studies**: In the ablation study, we provide incremental comparisons. Initially, we used averaged text embeddings and OOD probability as the baseline. We then added ID probability and PMM (Gaussian and Weibull distributions) to demonstrate the contribution of each component. This ablation study structure is consistent with that of PROB and CAT.
>
> We have included additional comparison experiments involving linear interpolation (LI) and sliding window (SW), as detailed in Table 2 of the attached document. The results indicate that the use of LI or SW individually does not lead to significant performance improvements. Both methods demonstrate only marginal enhancements over the original approach, indicating that neither LI nor SW can accurately model the data distribution when used in isolation. This is further substantiated by the fitting performance illustrated in Figure 2 (the attached document). The combined use of LI and SW yields the most substantial performance gains and achieves a more accurate fit to the original data distribution.
>
> Regarding the different text attribute designs you mentioned, using different LLMs could lead to unfair comparisons. Although we could utilize the latest LLMs like GPT-4o to generate text attributes or even combine multiple LLMs to generate richer text attributes, doing so would create an imbalance when comparing with models like FOMO and OWL-Vit. Therefore, we did not modify other settings to ensure fairness. Additionally, shift & scale is the default setting of the benchmark model. To align the output of OWL-Vit, we adopted the pre-trained shift & scale to rescale the similarity.
>
> **Content Discrepancies**: We are grateful for your observation regarding the mismatches between Figure 2 and the method descriptions, as well as the presence of typos. We sincerely apologize for these errors, which occurred due to inconsistencies between the final version of the manuscript and the initial structure. In the original version, we placed the problem definitions in Appendix A.1 as a separate section following the related work. However, as the detailed method descriptions took up more space than anticipated, we moved this section to the appendix, resulting in misaligned section indices in Figure 2. We have since conducted a thorough review of the entire manuscript, correcting all mismatches and typing errors. Regarding the algorithm's process, we have provided explanations in both Section 3 and Figure 2. Additionally, pseudocode implementation is provided at the beginning of the appendix, which we hope will help clarify any remaining questions.
>
> We hope our explanations help address your concerns and clarify our research strategies and decisions. Please reach out if you have further suggestions or need more information. Thank you again for your valuable feedback.

---

> > ### Comment · Reviewer_sXkA · 2024-08-12
> >
> > Thank you for the clarifications, which have addressed my concerns. To prevent potential confusion about the task setting, the authors are suggested to include the aforementioned differences with related tasks, such as OSOD and OVC, in the final version. I would like to raise my score to weak accept.

---

### Official Review · Reviewer_jKzy · 2024-07-09

**Soundness:** 3
**Presentation:** 3
**Contribution:** 2
**Rating:** 5
**Confidence:** 4

**Summary:**

This paper aims to understand the model’s behavior in predicting the unknown category.  First, the authors model the text attribute and the positive sample probability, obtaining their empirical probability, which can be seen as the detector’s estimation of the likelihood of the target with certain known attributes being predicted as the foreground. Then, they jointly decide whether the current object should be categorized in the unknown category based on the empirical, the in-distribution, and the out-of-distribution probability. Finally, based on the decision making process, the authors can infer the similarity of an unknown object to known classes and identify the attribute with the most significant impact on the decision-making process.

**Strengths:**

1. The paper is written well and is easy to understand.

2. The studied problem is very important.

3. The results seem to outperform state-of-the-art.

**Weaknesses:**

1. Is it able to utilize more effective OOD score for equations 13-14?

2. It might be more comprehensive to report more metrics, such as AUROC and FPR.

3. It might be useful to include ablation results on prompt templates and LLM used.

**Questions:**

see above

---

> ### Author Rebuttal · Authors · 2024-08-03
>
> We sincerely appreciate your valuable feedback on our work, particularly regarding the concerns related to the detailed technical design. Below is our detailed response:
>
> **OOD Score**: To detect unknown objects, we proposed a method that models the probability of an object being predicted as foreground. By modeling attribute similarity and the positive sample probability of the object, we generate the empirical probability (Empirical Prob). By combining the Empirical Prob with in-distribution probability (ID Prob), we can ensure the possibility that each sample will be recognized as a foreground object. Based on this, UMB can extract potential objects of interest from the background and label them as unknown. However, there is a challenge in distinguishing between known and unknown categories. The ID Prob is used for linear combination to determine the probability of known categories, while the Empirical Prob is also derived based on the confidence of known categories. As a result, both the ID and Empirical Prob are higher for a known object. This is why we introduce the out-of-distribution probability (OOD Prob). However, in our paper, we implemented a simple OOD Prob measure that assesses the uncertainty of known categories, specifically the probability that it belongs to the background. Admittedly, a more complex and representative probability score could be developed to calculate the current object's background probability. However, this would introduce additional computational overhead and complicate the UMB pipline. Therefore, we opted for the simplest method to measure the background probability, which can be accomplished with two matrix operations, ensuring the algorithm's simplicity and efficiency. These settings allowed UMB to achieve significant performance improvements in detecting both known and unknown categories, with a 5.3 mAP increase, establishing a new SOTA.
>
> **More Metrics**: In Table 1 of the attached PDF, we provide a comparison of additional metrics. We included the early OWOD (Open-World Object Detection) metrics used to evaluate model performance in unknown categories (WI, A-OSE, and Recall). WI assesses the difference in recognition accuracy between known and unknown categories, while A-OSE measures the number of errors where the model mistakenly classifies unknown objects as known categories. Our method still achieved leading performance, such as a consistently lower A-OSE compared to FOMO, while also achieving a higher recall.  The AUROC and FPR you mentioned are more commonly used to evaluate classification tasks. Object detection, however, consists of both classification and localization tasks.  Typically, a detector's post-processing filters out all regions predicted as negative samples, meaning that true negatives (TN) are not present in the prediction results. Therefore, metrics that rely on TN, such as FPR, cannot be computed. Thus, the evaluation of detection tasks generally relies on the results predicted as positive samples by the detector, such as recall and precision. However, both early OWOD evaluation metrics and individual recall or precision metrics have their limitations. Relying solely on recall can lead to a detector greedily predicting all possible regions as positive samples, while relying solely on precision can make the detector overly conservative. The early OWOD metrics emerged because detectors performed poorly on unknown objects, focusing on evaluating specific aspects of performance in unknown categories, such as A-OSE. However, with the introduction of open-vocabulary object detection models, detectors' performance on unknown categories has improved. Therefore, we adopted mAP, the most widely accepted metric for object detection, as it balances precision and recall and evaluates the detector's performance across different thresholds. This is also why mAP is typically used to evaluate detector performance in closed-set object detection. We understand that specific scenarios might require additional metrics, so we provided TP for each dataset, which, combined with other metrics, can be used to calculate other values, such as FP. We will include this table as part of the appendix to provide additional information. Additionally, we will release the training and validation settings and final weights to support the application and development of the OWOD.
>
> **Additional Experiments**: Prompt templates and the LLM used are crucial in our detector. For instance, using more templates to generate visual embeddings and utilizing different LLMs could achieve higher performance; however, this would lead to an unfair comparison with FOMO or OWL-Vit. Therefore, to prevent such disparities, we adopted consistent configurations during the training and inference of UMB to ensure fairness. If you are interested in further experiments, we have provided more detailed information on related experiments in the appendix.
>
> The above is our response to the concerns you raised regarding the detailed technical design. We hope that these explanations can clearly answer your questions and help you better understand the strategies and decisions we adopted in our research. We again express our gratitude for your valuable feedback. If you have any further suggestions or require additional information, please feel free to contact us.

---

### Official Review · Reviewer_bGxY · 2024-07-11

**Soundness:** 3
**Presentation:** 3
**Contribution:** 3
**Rating:** 8
**Confidence:** 3

**Summary:**

This paper proposes a new solution for the challenging task of Open World Object Detection (OWOD) by exploring the reasons behind non-classification and then using the textual attributes of unknown objects to infer the most similar known category. Evaluation results on multiple real-world application datasets show significant performance improvements.

**Strengths:**

A new approach to solving classification problems in Open World Object Detection, and verifying its effectiveness.

**Weaknesses:**

The writing of some mathematical symbols needs to be improved.

**Questions:**

(1)The limitation discussion is a bit insufficient. It could be better to add some visualizations of the failure examples
(2)Some mathematical variable symbols(e.g., Eq.6 and Eq.7 ) are not standard. It is recommended that more common and standard notations be used.

**Limitations:**

Yes

---

> ### Author Rebuttal · Authors · 2024-08-01
>
> We thank the reviewer for their constructive feedback, which will help us improve the quality of our manuscript. Below, we address each of the points raised.
>
> **Limitation Discussion**: We appreciate the reviewer's suggestion to enhance the limitation discussion with visualizations of failure examples. We agree that visual representations can provide clearer insights into the nature of the limitations. In the revised manuscript, we will include several visualizations illustrating common failure modes encountered during our experiments. These visual examples will be included in the Appendix of the paper, as illustrated in the attached PDF (Figure 1). We present typical cases of detection failures from each dataset, focusing on recall capability and detection accuracy.
>
> For instance, in the Aquatic dataset, UMB failed to detect small orange fish, while in the Aerial dataset, it did not successfully recall vehicles. These instances reveal the detector’s shortcomings in recall capability. Moreover, in the Game and Surgery datasets, UMB displayed occurrences of repeated predictions. Nevertheless, UMB still outperformed FOMO in overall performance. Specifically, in the Aquatic dataset, UMB accurately located the contours of the fish, whereas FOMO showed deviations in contour localization and even missed similar objects. Furthermore, FOMO incorrectly identified the reflection of the photographer’s shoes in the glass as an unknown object, demonstrating lower precision. Similar issues were observed in the Aerial and Game datasets, where FOMO often confused objects with the background, resulting in new erroneous predictions, such as misidentifying rooftops as a single object in the Aerial dataset. However, UMB did not commit the same errors in these cases. In summary, although UMB also exhibited some false detections in certain scenarios, it outperformed FOMO in both detection accuracy and the ability to recall potential objects.
>
> These visualizations will highlight specific instances where the proposed method did not perform as expected, along with a brief analysis of each case. By doing so, we aim to provide a more comprehensive understanding of the limitations and the contexts in which they occur.
>
> **Mathematical Variable Symbols:** We appreciate the reviewer's attention to the mathematical notation used in our paper. To improve clarity and consistency, we will revise the mathematical symbols in Equations 6 and 7 to align with more commonly accepted and standard notations. This will ensure that the equations are easily understood by a broader audience and adhere to conventional mathematical standards.
>
> Thank you once again for your valuable feedback. If you have further suggestions or need more information support, please feel free to contact us. Thank you again for your recognition and valuable feedback on our work.

---

> > ### Comment · Reviewer_bGxY · 2024-08-12
> >
> > Thank the authors for their response, which addressed my concerns. I am keeping my final rating at 8: Strong Accept.

---

### Official Review · Reviewer_N6Np · 2024-07-12

**Soundness:** 3
**Presentation:** 2
**Contribution:** 3
**Rating:** 6
**Confidence:** 4

**Summary:**

This paper introduces a new open-world object detection model (UMB) aimed at understanding the model's behavior when predicting unknown categories. By modeling text attributes and positive sample probability, the paper proposes a joint decision-making method based on empirical probability, in-distribution probability, and out-of-distribution probability to infer the similarity of unknown objects to known classes and identify the attributes that have the most significant impact on the decision-making process. The evaluation results on the Real-World Object Detection (RWD) benchmark show that this method surpasses the previous state-of-the-art (SOTA) with an absolute gain of 5.3 mAP for unknown classes, reaching 20.5 mAP.

**Strengths:**

First to focus on understanding the model's behavior in unknown category predictions.
Proposes a new framework (UMB) capable of detecting unknown categories and understanding model behavior using textual descriptions.
Shows significant performance improvements on the Real-World Object Detection (RWD) benchmark, especially in unknown categories.

**Weaknesses:**

1.	The method relies on high-quality text attributes and positive sample probability modeling. Performance may be affected if data quality is poor or attributes are incomplete.
2.	While performing well on the RWD dataset, its generalization ability to other domains and datasets needs further verification, possibly requiring more testing in different environments
3.	The robustness of using the most influencing attributes in the decision process to infer the similarity between unknown and known objects needs further validation.
4.	For large models, the performance and quality requirements for samples and attributes are high. The categories, quantity, and quality of samples and attributes significantly affect network performance and need further analysis.

**Questions:**

What are the computational resource requirements for this method in practical applications? Is the method feasible in resource-constrained environments?
Can this method maintain the same performance improvements on other datasets and in other domains? Further validation in more real-world environments is needed to ensure its broad applicability.
How robust is the use of the most influencing attributes in inferring the similarity between unknown and known objects? Could noise or outlier data affect the decision-making process?

**Limitations:**

The method is quite complex and may require substantial computational resources and time for training and inference. This could limit its use in practical applications with limited computational resources.

---

> ### Author Rebuttal · Authors · 2024-07-31
>
> We greatly appreciate your valuable comments on our work, particularly your concerns about similarity verification, resource consumption, and deployment. Here are our detailed responses:
>
> **Similarity Verification**. Ideally, we would like to annotate every unknown object in the RWD dataset and identify its most similar known class. However, this approach presents several challenges: 1) Workload issue: A single image may contain multiple unknown objects, such as stones, making it impractical to annotate each unknown object and find the most similar known class in practice. 2) Subjectivity and inconsistency: When manually annotating the similarity between unknown objects and known classes, subjective judgments are inevitable. These can lead to differing opinions among annotators regarding the similarity of the same unknown object to known classes, resulting in inconsistent annotations. Moreover, the diversity and complexity of unknown objects make it difficult to establish a unified annotation standard. Therefore, manual evaluation of similarity is not only unfeasible in workload but also difficult to ensure the objectivity and consistency.
>
> In addition, our evaluation focuses on how the detector determines the similarity between unknown objects and known classes during the decision-making process, which reflects the detector’s understanding of the current object. In subsequent improvements, we can use this to customize the distribution of the dataset. For example, when the detector considers that unknown object A is similar to known class B, and we want to intervene in this result, we can add some samples that are not similar to the known class B but belong to unknown object A through incremental learning to urge the detector to distinguish them. This is why we propose the concept of the unknown object most similar to the known class.
>
> Despite the challenges of manual annotation, we provide an approximate evaluation in Appendix (A.7). We found that the detector may produce duplicate predictions when predicting unknown objects. These duplicate predictions reflect the detector’s confusion between the current unknown object and known classes. Therefore, we collected all duplicate predictions and calculated the overlap between these duplicate predictions and the known classes given by the detector to indirectly evaluate the accuracy. Specifically, we calculated the IOU of all known and unknown predictions. If the IOU is greater than 0.95 (the highest value of the COCO standard), the current known and unknown prediction are considered overlapping. In the left figure, we show the number of these duplicate predictions, ending with _Repeat. Subsequently, we calculated the number of matches in these duplicate predictions, ending with _matched, indicating the cases where the most similar known class predicted by the detector is consistent with the overlapping class. In the right figure, we show the corresponding accuracy. From the experimental results, the accuracy reached about 90%. This indirect evaluation proves that UMB has high accuracy in predicting the unknown object most similar to the known class.
>
> **Resource consumption**.  We provided them in Appendix (A.4). All experiments were conducted a single NVIDIA GeForce RTX 4090 GPU. Since OWL-Vit has made all weights public, we can download these weights as a starting point and freeze them. Subsequently, we use text encoder to encode all sentences into text embeddings, and training these text embeddings for known class detection. To detect unknown objects, we only introduce a module (TAM) that needs to be trained. The training of TAM is lightweight, and it only needs to train up to 5 groups of probability models. For example, using the Gaussian model, the number of parameters to be trained is 10, including 5 variances and 5 means. This is why we can complete all experiments on a single NVIDIA GeForce RTX 4090. For the inference, compared to the original OWL-Vit, we only introduced the calculation of OOD (Out-of-Distribution), ID (In-Distribution), and Empirical probability. However, these three parts of the calculation can be completed through a few matrix operations. Compared with the inference computation of OWL-Vit, the increase in computation is negligible. Therefore, our inference resource consumption is almost consistent with OWL-Vit.
>
> **Deployment**. The deployment of object detection is a very complex issue, which needs to consider the restrictions of different devices and environments. At present, the most commonly used deployment scheme is to convert the original PyTorch model into a TensorRT model, and use the optimization acceleration of the TensorRT model (such as FP16 or INT8 inference) to achieve real-time inference. When deploying the UMB model, all texts can be preprocessed into text embeddings, and the text encoder can be discarded, and only the visual encoder is finally deployed on the device. This makes the deployment scheme consistent with OWL-Vit. The newly introduced probability calculation part follows the standard PyTorch implementation, so it can also be converted into a TensorRT model. As we mentioned in inference resource consumption, the computational burden brought by our method is negligible. Therefore, our method can run normally on all devices where OWL-Vit models are applicable. In addition, our method can also serve as a pluggable module, migrated to other OVC-style detectors (e.g. YOLO-World), giving them the ability to detect unknown objects and understand the decision-making process.
>
> The above are our responses to similarity verification, resource consumption, and deployment. We hope that these explanations can clearly answer your questions and help you better understand the strategies and decisions we adopted in our research. If you have further suggestions or need more information support, please feel free to contact us. Thank you again for your recognition and valuable feedback on our work.

---

> > ### Comment · Reviewer_N6Np · 2024-08-12
> >
> > Thank you for the author's response, which solved most of my problems. I maintain my final rating: 6 Weak Accept.

---

### Author Rebuttal · Authors · 2024-08-03

We sincerely appreciate your valuable feedback, which has been instrumental in guiding us to improve our work. In the attached document, we have provided additional experiments, including examples of detection failures (Figure 1), a comparison of more metrics (Table 1), further ablation studies (Table 2), and the corresponding fitting visualizations (Figure 2). We hope this additional information will help you better understand the content conveyed and the technical design implemented in the paper. Thank you once again for your feedback. If you have any further questions or require additional clarification, please do not hesitate to contact us.

---

### Decision · Program_Chairs · 2024-09-25

**Decision:**

Accept (poster)

**Comment:**

This work attempts to understand the model’s prediction when identifying unlabeled objects and establish connections between unknown and known categories. All reviewers rate positively. During the rebuttal & discussion process, concerns, like evaluation metrics and lack of some ablation studies, have been solved effectively. After careful consideration, The AC is inclined to accept this paper.